# Regret Bounds for Information-Directed Reinforcement Learning

**Botao Hao**
Deepmind
`haobotao000@gmail.com`

**Tor Lattimore**
Deepmind
`lattimore@google.com`

## Abstract

Information-directed sampling (IDS) has revealed its potential as a data-efficient algorithm [Lu et al., 2021] for reinforcement learning (RL). However, theoretical understanding of IDS for Markov Decision Processes (MDPs) is still limited. We develop novel information-theoretic tools to bound the information ratio and cumulative information gain about the learning target. Our theoretical results shed light on the importance of choosing the learning target such that the practitioners can balance the computation and regret bounds. As a consequence, we derive prior-free Bayesian regret bounds for `vanilla-IDS` which learns the whole environment under tabular finite-horizon MDPs. In addition, we propose a computationally-efficient `regularized-IDS` that maximizes an additive form rather than the ratio form and show that it enjoys the same regret bound as `vanilla-IDS`. With the aid of rate-distortion theory, we improve the regret bound by learning a surrogate, less informative environment. Furthermore, we extend our analysis to linear MDPs and prove similar regret bounds for Thompson sampling as a by-product.

## 1   Introduction

Information-directed sampling (IDS) is a *design principle* proposed by [Russo and Van Roy, 2014, 2018] that optimizes the trade-off between *information* and *regret*. Comparing with other design principles such as UCB and Thompson sampling (TS), IDS can automatically adapt to different information-regret structures. As a result, IDS demonstrates impressive empirical performance [Russo and Van Roy, 2018] and outperforms UCB and TS in terms of asymptotic optimality [Kirschner et al., 2021] and minimax optimality in heteroscedastic bandits [Kirschner and Krause, 2018] and sparse linear bandits [Hao et al., 2021].

In the context of full RL, mutiple works have examined the empirical performance of IDS [Nikolov et al., 2018, Lu et al., 2021]. However, formal regret guarantee for IDS is still lacking. IDS minimizes a notion of *information ratio* that is the ratio of per-episode regret and information gain about the learning target. While different choices of the learning target could lead to different regret bounds and computational methods, the most natural choice is the whole environment and we name the corresponding IDS as `vanilla-IDS`.

In this work, we prove the first prior-free $\widetilde{O}(\sqrt{S^3 A^2 H^4 L})$ Bayesian regret bound for `vanilla-IDS`, where $S$ is the size of state space, $A$ is the size of action space, $H$ is the length of episodes and $L$ is the number of episodes. Computationally, `vanilla-IDS` needs to optimize over the full policy space, which is not efficient in general. To facilitate the computation, we consider its regularized form, named `regularized-IDS`, that can be solved by any dynamic programming solver. By carefully

choosing the tunable parameter, we prove that `regularized-IDS` enjoys the same regret bound as `vanilla-IDS`.

Although learning the whole environment offers certain computational advantages, the agent could take too much information to learn the whole environment exactly. A key observation is that different states may correspond to the same value function which eventually determines the behavior of the optimal policy. Through the rate-distortion theory, we construct a surrogate environment that is less informative to learn but enough to identify the optimal policy. As a result, we propose `surrogate-IDS` that takes the surrogate environment as the learning target and prove a sharper $\widetilde{O}(\sqrt{S^2 A^2 H^4 L})$ bound for tabular MDPs.

In the end, we extend our analysis to linear MDPs where we *must* learn a surrogate environment due to potentially infinitely many states and derive a $\widetilde{O}(dH^2\sqrt{T})$ Bayesian regret bound that matches the existing minimax lower bound up to a factor of $H$. As a by-product of our analysis, we also prove prior-free Bayesian regret bounds for TS under tabular and linear MDPs.

## 2 Related work

In general, there are two ways to prove Bayesian regret bounds. The first is to introduce confidence sets such that the Bayesian regret bounds of TS can match the best possible frequentist regret bounds by UCB [Russo and Van Roy, 2014] and has been extended to RL by Osband et al. [2013], Osband and Van Roy [2014], Osband et al. [2019]. However, when the best possible bound for UCB is sub-optimal (for instance, sparse linear bandits [Hao et al., 2021]), this technique yields a sub-optimal Bayesian regret bound. In addition, this technique can only be used to analyze TS but not IDS.

The second is to decompose the Bayesian regret into an *information ratio* term and a *cumulative information gain* term and bound them by tools from information theory [Russo and Van Roy, 2016]. This technique can be used to analyze both TS [Dong and Van Roy, 2018, Bubeck and Sellke, 2020] and IDS in bandits setting [Russo and Van Roy, 2014, Liu et al., 2018, Kirschner et al., 2020b, Hao et al., 2021, 2022], partial monitoring [Lattimore and Szepesvári, 2019, Kirschner et al., 2020a, Lattimore and Gyorgy, 2021] but not in RL as far as we know. One exception is Lu and Van Roy [2019], Lu [2020] who bounded the information ratio for a specific Dirichlet prior with additional assumptions.

Frequentist regret bounds in episodic RL have received considerable attention recently. For tabular MDPs, several representative works include UCBVI [Azar et al., 2017], optimistic Q-learning [Jin et al., 2018], RLSVI [Russo, 2019], UCB-Advantage [Zhang et al., 2020], UCB-MQ [Ménard et al., 2021]. While our regret bounds are not state of the art, the primary goal of this paper is to broaden the set of efficient RL design principles known to satisfy $\sqrt{T}$ regret bounds.

For linear or linear mixture MDPs, several representative works include LSVI-UCB [Jin et al., 2020], OPPO [Cai et al., 2020], UCRL-VTR [Ayoub et al., 2020, Zhou et al., 2021], RLSVI [Zanette et al., 2020]. Notably, Zhang [2021], Dann et al. [2021] derived minimax regret bounds for a variant of TS. Beyond linear cases, several works consider general function approximation based on Bellman rank [Jiang et al., 2017], eluder dimension [Wang et al., 2020], Bellman-eluder dimension [Jin et al., 2021] and bilinear class [Du et al., 2021].

It is worth mentioning the recent impressive work by Foster et al. [2021] who proposed a general Estimation-to-Decisions (E2D) design principle. Although motivated by different design principles, E2D shares the similar form as `regularized-IDS`. On one hand, Foster et al. [2021] mainly focuses on statistical complexity in a minimax sense, while we offer a specific computationally-efficient algorithm thanks to the chain rule of mutual information and independent priors and derive corresponding Bayesian regret bounds. On the other hand, while E2D tends to learn the whole environment, our theory in Section 5 suggests learning a surrogate environment could yield better regret bounds.

# 3 Preliminary

**Finite-horizon MDPs** The environment is characterized by a finite-horizon time-inhomogeneous MDP, which is a tuple $\mathcal{E} = (\mathcal{S}, \mathcal{A}, H, \{P_h\}_{h=1}^H, \{r_h\}_{h=1}^H)$, where $\mathcal{S}$ is the countable state space with $|\mathcal{S}| = S$, $\mathcal{A}$ is the finite action space with $|\mathcal{A}| = A$, $H$ is the episode length, $P_h : \mathcal{S} \times \mathcal{A} \to \Delta_{\mathcal{S}}$ is the transition probability kernel and $r_h : \mathcal{S} \times \mathcal{A} \to [0,1]$ is the reward function. For a finite set $\mathcal{S}$, let $\Delta_{\mathcal{S}}$ be the set of probability distributions over $\mathcal{S}$. We assume $\mathcal{S}, \mathcal{A}, r_h$ are known and deterministic while the transition probability kernel is unknown and random. Throughout the paper, we may write $P_h$ and $r_h$ explicitly depend on $\mathcal{E}$ when necessary.

Let $\Theta_h = [0,1]^{S \times A \times S}$ be the parameter space of $P_h$ and $\Theta = \Theta_1 \times \cdots \times \Theta_H$ be the full parameter space. We assume $\rho_h$ is the prior probability measure for $P_h$ on $\Theta_h$ with Borel $\sigma$-algebra and $\rho = \rho_1 \otimes \cdots \otimes \rho_H$ as the product prior probability measure for the whole environment on $\Theta$ with Borel $\sigma$-algebra. This ensures the priors over different layers are independent and the prior is assumed to be known to the learner.

**Interaction protocol** An agent interacts with a finite-horizon MDP as follows. The initial state $s_1^\ell$ is assumed to be fixed over episodes. In each episode $\ell \in [L]$ and each layer $h \in [H]$, the agent observes a state $s_h^\ell$, takes an action $a_h^\ell$, and receives a reward $r_h^\ell$. Then, the environment evolves to a random next state $s_{h+1}^\ell$ according to distribution $P_h(\cdot|s_h^\ell, a_h^\ell)$. The episode terminates when $s_{H+1}$ is reached and is reset to the initial state.

Denote $\mathcal{H}_{\ell,h}$ as the history of episode $\ell$ up to layer $h$, e.g., $\mathcal{H}_{\ell,h} = (s_1^\ell, a_1^\ell, r_1^\ell, \ldots, s_h^\ell, a_h^\ell, r_h^\ell)$ and the set of such possible history is $\Omega_h = \prod_{i=1}^h (\mathcal{S} \times \mathcal{A} \times [0,1])$. Let $\mathcal{D}_\ell = (\mathcal{H}_{1,H}, \ldots, \mathcal{H}_{\ell-1,H})$ as the entire history up to episode $\ell$ with $\mathcal{D}_1 = \emptyset$. A policy $\pi$ is a collection of (possibly randomised) mappings $(\pi_1, \ldots, \pi_H)$ where each $\pi_h$ maps an element from $\Omega_{h-1} \times \mathcal{S}$ to $\Delta(\mathcal{A})$ and $\Pi$ is the whole policy class. A *stationary policy* chooses actions based on only the current state and current layer. The set of such policies is denoted by $\Pi_S$ where we denote $\pi_h(a|s)$ as the probability that the agent chooses action $a$ at state $s$ and layer $h$.

**Value function** For each $h \in [H]$ and a policy $\pi$, the value function $V_{h,\pi}^{\mathcal{E}} : \mathcal{S} \to \mathbb{R}$ is defined as the expected value of cumulative rewards received under policy $\pi$ when starting from an arbitrary state at $h$th layer; that is,

$$V_{h,\pi}^{\mathcal{E}}(s) := \mathbb{E}_\pi^{\mathcal{E}} \left[ \sum_{h'=h}^H r_{h'}(s_{h'}, a_{h'}) \middle| s_h = s \right],$$

where $\mathbb{E}_\pi^{\mathcal{E}}$ denotes the expectation over the sample path generated under policy $\pi$ and environment $\mathcal{E}$. We adapt the convention that $V_{H+1,\pi}^{\mathcal{E}}(\cdot) = 0$. There always exists an optimal policy $\pi^*$ which gives the optimal value $V_{h,\pi^*}^{\mathcal{E}}(s) = \max_{\pi \in \Pi_S} V_{h,\pi}^{\mathcal{E}}(s)$ for all $s \in \mathcal{S}$ and $h \in [H]$. Note that in the Bayesian setting, $\pi^*$ is a function of $\mathcal{E}$ so it is also a random variable. In addition, we define the action-value function as follows:

$$Q_{h,\pi}^{\mathcal{E}}(s,a) := \mathbb{E}_\pi^{\mathcal{E}} \left[ \sum_{h'=h}^H r_{h'}(s_{h'}, a_{h'}) \middle| s_h = s, a_h = a \right],$$

which satisfies the Bellman equation: $Q_{h,\pi}^{\mathcal{E}}(s,a) = r_h(s,a) + \mathbb{E}_{s' \sim P_h(\cdot|s,a)}[V_{h+1,\pi}^{\mathcal{E}}(s')]$. Furthermore, we denote the *state-action occupancy measure* as

$$d_{h,\pi}^{\mathcal{E}}(s,a) = \mathbb{P}_\pi^{\mathcal{E}}(s_h = s, a_h = a),$$

where we denote $\mathbb{P}_\pi^{\mathcal{E}}$ as the law of the sample path generated under policy $\pi$ and environment $\mathcal{E}$.

**Bayesian regret** The agent interacts with the environment for $L$ episodes and the total number of steps is $T = LH$. The expected cumulative regret of an algorithm $\pi = \{\pi^\ell\}_{\ell=1}^L$ with respect to an environment $\mathcal{E}$ is defined as

$$\mathfrak{R}_L(\mathcal{E}, \pi) = \mathbb{E} \left[ \sum_{\ell=1}^L \left( V_{1,\pi^*}^{\mathcal{E}}(s_1^\ell) - V_{1,\pi^\ell}^{\mathcal{E}}(s_1^\ell) \right) \right],$$

where the expectation is taken with respect to the randomness of $\pi^\ell$. The Bayesian regret then is defined as

$$\mathfrak{BR}_L(\pi) = \mathbb{E}[\mathfrak{R}_L(\mathcal{E}, \pi)],$$

where the expectation is taken with respect to the prior distribution of $\mathcal{E}$. At each episode, TS finds

$$\pi_{\text{TS}}^\ell = \underset{\pi \in \Pi}{\operatorname{argmax}} \, V_{1,\pi}^{\mathcal{E}_\ell}(s_1^\ell),$$

where $\mathcal{E}_\ell$ is a sample from the posterior distribution of $\mathcal{E}$, e.g., $\mathcal{E}_\ell \sim \mathbb{P}(\mathcal{E} \in \cdot | \mathcal{D}_\ell)$.

**Notations**  Let $(\Omega, \mathcal{F}, \mathbb{P})$ as a measurable space. A random variable $X$ is a measureable function $X : \Omega \to E$ from a set of possible outcomes $\Omega$ to a measurable space $E$. Now $\mathbb{P}(X \in \cdot)$ is a probability measure that maps from $\mathcal{F}$ to $[0, 1]$. $\mathcal{D}_\ell$ is another random variable from $\Omega$ to a measurable space $Y$. Then $\mathbb{P}(X \in \cdot | \mathcal{D}_\ell)$ is a probability kernel that maps from $\Omega \times \mathcal{F} \to [0, 1]$.

We write $\mathbb{P}_\ell(\cdot) = \mathbb{P}(\cdot | \mathcal{D}_\ell)$, $\mathbb{E}_\ell[\cdot] = \mathbb{E}[\cdot | \mathcal{D}_\ell]$ and also define the conditional mutual information $\mathbb{I}_\ell(X; Y) = D_{\text{KL}}(\mathbb{P}((X, Y) \in \cdot | \mathcal{D}_\ell) || \mathbb{P}(X \in \cdot | \mathcal{D}_\ell) \otimes \mathbb{P}(Y \in \cdot | \mathcal{D}_\ell))$. For a random variable $\chi$ we define:

$$\mathbb{I}_\ell^\pi(\chi; \mathcal{H}_{\ell,h}) = D_{\text{KL}}(\mathbb{P}_{\ell,\pi}((\chi, \mathcal{H}_{\ell,h}) \in \cdot) || \mathbb{P}_{\ell,\pi}(\chi \in \cdot) \otimes \mathbb{P}_{\ell,\pi}(\mathcal{H}_{\ell,h} \in \cdot)),$$

where $\mathbb{P}_{\ell,\pi}$ is the law of $\chi$ and the history induced by policy $\pi$ interacting with a sample from the posterior distribution of $\mathcal{E}$ given $\mathcal{D}_\ell$. We define $\bar{\mathcal{E}}_\ell$ as the mean MDP where for each state-action pair $(s, a)$, $P_h^{\bar{\mathcal{E}}_\ell}(\cdot | s, a) = \mathbb{E}_\ell[P_h^{\mathcal{E}}(\cdot | s, a)]$ is the mean of posterior measure.

# 4  Learning the whole environment

The core design of IDS for RL relies on a notion of *information ratio*. The information ratio for a policy $\pi$ at episode $\ell$ is defined as

$$\Gamma_\ell(\pi, \chi) := \frac{(\mathbb{E}_\ell[V_{1,\pi^*}^{\mathcal{E}}(s_1^\ell) - V_{1,\pi}^{\mathcal{E}}(s_1^\ell)])^2}{\mathbb{I}_\ell^\pi(\chi; \mathcal{H}_{\ell,H})}, \tag{4.1}$$

where $\chi$ is the learning target to prioritize information sought by the agent. The choice of $\chi$ plays a crucial role in designing the IDS and could lead to different regret bounds and computational methods. We first consider the most natural choice of $\chi$ which is the whole environment $\mathcal{E}$.

## 4.1  Vanilla IDS

`Vanilla-IDS` takes the whole environment $\mathcal{E}$ as the learning target and at the beginning of each episode, the agent computes a stochastic policy:

$$\pi_{\text{IDS}}^\ell = \underset{\pi \in \Pi}{\operatorname{argmin}} \left[ \Gamma_\ell(\pi) := \frac{(\mathbb{E}_\ell[V_{1,\pi^*}^{\mathcal{E}}(s_1^\ell) - V_{1,\pi}^{\mathcal{E}}(s_1^\ell)])^2}{\mathbb{I}_\ell^\pi(\mathcal{E}; \mathcal{H}_{\ell,H})} \right]. \tag{4.2}$$

Define the worst-case information ratio $\Gamma^*$ such that $\Gamma_\ell(\pi_{\text{IDS}}^\ell) \leq \Gamma^*$ for any $\ell \in [L]$ almost surely. The next theorem derives a generic regret bound for `vanilla-IDS` in terms of $\Gamma^*$ and the mutual information between $\mathcal{E}$ and the history.

**Theorem 4.1.** *A generic regret bound for* `vanilla-IDS` *is*

$$\mathfrak{BR}_L(\pi_{IDS}) \leq \sqrt{\mathbb{E}[\Gamma^*] \mathbb{I}(\mathcal{E}; \mathcal{D}_{L+1}) L}.$$

The proof is deferred to Appendix A.1 and follows standard information-theoretical regret decomposition and the chain rule of mutual information that originally was exploited by Russo and Van Roy [2014]. For tabular MDPs, it remains to bound the $\mathbb{E}[\Gamma^*]$ and $\mathbb{I}(\mathcal{E}; \mathcal{D}_{L+1})$ separately.

**Lemma 4.2.** *The worst-case information ratio for tabular MDPs is upper bounded by*

$$\mathbb{E}[\Gamma^*] \leq 2SAH^3.$$

We sketch the main steps of the proof and defer the full proof to Appendix A.2.

*Proof sketch.* Since `vanilla-IDS` minimizes the information ratio over all the policies, we can bound the information ratio of `vanilla-IDS` by the information ratio of TS.

- *Step one.* Our regret decomposition uses the value function based on $\bar{\mathcal{E}}_\ell$ as a bridge:

$$\mathbb{E}_\ell\left[V_{1,\pi^*}^{\mathcal{E}}(s_1^\ell) - V_{1,\pi_{\mathrm{TS}}^\ell}^{\mathcal{E}}(s_1^\ell)\right] = \underbrace{\mathbb{E}_\ell\left[V_{1,\pi^*}^{\mathcal{E}}(s_1^\ell) - V_{1,\pi_{\mathrm{TS}}^\ell}^{\bar{\mathcal{E}}_\ell}(s_1^\ell)\right]}_{I_1} + \underbrace{\mathbb{E}_\ell\left[V_{1,\pi_{\mathrm{TS}}^\ell}^{\bar{\mathcal{E}}_\ell}(s_1^\ell) - V_{1,\pi_{\mathrm{TS}}^\ell}^{\mathcal{E}}(s_1^\ell)\right]}_{I_2}.$$

  Note that conditional on $\mathcal{D}_\ell$, the law of $\pi_{\mathrm{TS}}^\ell$ is the same as the law of $\pi^*$ and both $\pi^*$ and $\pi_{\mathrm{TS}}^\ell$ are independent of $\bar{\mathcal{E}}_\ell$. This implies $\mathbb{E}_\ell[V_{1,\pi_{\mathrm{TS}}^\ell}^{\bar{\mathcal{E}}_\ell}(s_1^\ell)] = \mathbb{E}_\ell[V_{1,\pi^*}^{\bar{\mathcal{E}}_\ell}(s_1^\ell)]$.

- *Step two.* Denote $\Delta_h^{\mathcal{E}}(s,a) = \mathbb{E}_{s'\sim P_h^{\mathcal{E}}(\cdot|s,a)}[V_{h+1,\pi^*}^{\mathcal{E}}(s')] - \mathbb{E}_{s'\sim P_h^{\bar{\mathcal{E}}}(\cdot|s,a)}[V_{h+1,\pi^*}^{\mathcal{E}}(s')]$ as the value function difference. Inspired by Foster et al. [2021], with the use of *state-action occupancy measure* and Lemma D.3, we can derive

$$I_1 = \sum_{h=1}^H \mathbb{E}_\ell\left[\sum_{(s,a)} \frac{d_{h,\pi^*}^{\bar{\mathcal{E}}_\ell}(s,a)}{(\mathbb{E}_\ell[d_{h,\pi^*}^{\bar{\mathcal{E}}_\ell}(s,a)])^{1/2}}(\mathbb{E}_\ell[d_{h,\pi^*}^{\bar{\mathcal{E}}_\ell}(s,a)])^{1/2}\Delta_h^{\mathcal{E}}(s,a)\right].$$

  Applying the Cauchy–Schwarz inequality and Pinsker's inequality (see Eqs. (A.2)-(A.4) in the appendix for details), we can obtain

$$I_1 \le \sqrt{SAH^3}\left(\sum_{h=1}^H \mathbb{E}_\ell\left[\mathbb{E}_{\pi_{\mathrm{TS}}^\ell}^{\bar{\mathcal{E}}_\ell}\left[\frac{1}{2}D_{\mathrm{KL}}\left(P_h^{\mathcal{E}}(\cdot|s_h^\ell,a_h^\ell)||P_h^{\bar{\mathcal{E}}_\ell}(\cdot|s_h^\ell,a_h^\ell)\right)\right]\right]\right)^{1/2},$$

  where we interchange $\pi_{\mathrm{TS}}^\ell$ and $\pi^*$ again and $\mathbb{E}_{\pi_{\mathrm{TS}}^\ell}^{\bar{\mathcal{E}}_\ell}$ is taken with respect to $s_h^\ell, a_h^\ell$ and $\mathbb{E}_\ell$ is taken with respect to $\pi_{\mathrm{TS}}^\ell$ and $\mathcal{E}$.

- *Step three.* It remains to establish the following equivalence of above KL-divergence and the information gain (Lemma A.1):

$$\sum_{h=1}^H \mathbb{E}_\ell\left[\mathbb{E}_{\pi_{\mathrm{TS}}^\ell}^{\bar{\mathcal{E}}_\ell}\left[D_{\mathrm{KL}}\left(P_h^{\mathcal{E}}(\cdot|s_h,a_h)||P_h^{\bar{\mathcal{E}}_\ell}(\cdot|s_h,a_h)\right)\right]\right] = \mathbb{I}_\ell^{\pi_{\mathrm{TS}}^\ell}\left(\mathcal{E};\mathcal{H}_{\ell,H}\right).$$

  A crucial step is to use the linearity of the expectation and the independence of priors over different layers (from the product prior as we assumed in Section 3) to show

$$\mathbb{P}_{\ell,\pi_{\mathrm{TS}}^\ell}(s_{h-1}=s, a_{h-1}=a) = \mathbb{P}_{\pi_{\mathrm{TS}}^\ell}^{\bar{\mathcal{E}}_\ell}(s_{h-1}=s, a_{h-1}=a).$$

Combining Steps 1-3, we can reach the conclusion and the bound for $I_2$ is similar. □

The next lemma directly bounds the mutual information for tabular MDPs.

**Lemma 4.3.** *The mutual information can be bounded by*

$$\mathbb{I}(\mathcal{E};\mathcal{D}_{L+1}) \le 2S^2 AH \log\left(SLH\right).$$

The proof relies on the construction of Bayes mixture density and a covering set for KL-divergence and is deferred to Appendix A.3. Combining Theorem 4.1, Lemmas 4.2 and 4.3 yields the following:

**Theorem 4.4** (Regret bound for tabular MDPs). *Suppose $\pi_{IDS} = \{\pi_{IDS}^\ell\}_{\ell=1}^L$ is the vanilla IDS policy. The following Bayesian regret bound holds for tabular MDPs*

$$\mathfrak{BR}_L(\pi_{IDS}) \le \sqrt{8S^3 A^2 H^4 L \log(SLH)}.$$

Although this regret bound is sub-optimal, this is the first sub-linear prior-free Bayesian regret bound for `vanilla-IDS`.

**Remark 4.5.** *It is worth mentioning that [Lu and Van Roy [2019], Lu [2020]](#) also derived Bayesian regret bound using information-theoretical tools but only hold for a specific Dirichlet prior as well other distribution-specific assumptions. Their proof heavily exploits the property of Dirichlet distribution and can not easily be extended to prior-free regret bounds.*

In the context of finite-horizon MDPs, [Lu et al. [2021]](#) considered a `conditional-IDS` such that at each time step, conditional on $s_h^\ell$, `conditional-IDS` takes the action according to

$$\pi_h(\cdot|s_h^\ell) = \underset{\nu \in \Delta_\mathcal{A}}{\mathrm{argmin}} \, \frac{\left(\mathbb{E}_\ell\left[V_{h,\pi^*}^\mathcal{E}(s_h^\ell) - Q_{h,\pi^*}^\mathcal{E}(s_h^\ell, A_h)\right]\right)^2}{\mathbb{I}_\ell\left(\chi; (A_h, Q_{h,\pi^*}^\mathcal{E}(s_h^\ell, A_h))\right)},$$

where $A_h$ is sampled from $\nu$. `Conditional-IDS` defined the information ratio *per-step* rather than *per-episode* such that it only needs to optimize over action space rather than the policy space. This offers great computational benefits but there is no regret guarantee for `conditional-IDS`. Recently, [Hao et al. [2022]](#) has demonstrated the theoretical limitation of `conditional-IDS` in contextual bandits.

## 4.2 Regularized IDS

Computing an IDS policy practically usually involves two steps: 1. *approximating the information ratio*; 2. *optimizing the information ratio*. In bandits where the optimal policy is only a function of action space, optimizing Eq. (4.2) is a convex optimization problem and has an optimal solution with at most two non-zero components ([Russo and Van Roy [2018](#), Proposition 6]). However in MDPs where the optimal policy is a mapping from the state space to the action space, `vanilla-IDS` needs to traverse two non-zero components over the full policy space which suggests the computational time might grow exponentially in $S$ and $H$.

To overcome this obstacle, we propose `regularized-IDS` that can be efficiently computed by any dynamic programming solver and enjoy the same regret bound as `vanilla-IDS`. At each episode $\ell$, `regularized-IDS` finds the policy:

$$\pi_{\text{r-IDS}}^\ell = \underset{\pi \in \Pi}{\mathrm{argmax}} \, \mathbb{E}_\ell[V_{1,\pi}^\mathcal{E}(s_1^\ell)] + \lambda \mathbb{I}_\ell\left(\mathcal{E}; \mathcal{H}_{\ell,H}^\pi\right), \tag{4.3}$$

where $\lambda > 0$ is a tunable parameter.

To approximate the objective function in Eq. (4.3), we assume the access to a *posterior sampling oracle*.

**Definition 4.6** (Posterior sampling oracle). *Given a prior over $\mathcal{E}$ and history $\mathcal{D}_\ell$, the posterior sampling oracle, SAMP, is a subroutine which returns a sample from the posterior distribution $\mathbb{P}_\ell(\mathcal{E})$. Multiple calls to the procedure result in independent samples.*

**Remark 4.7.** *SAMP can be exactly obtained when the conjugate prior such as Dirichlet distribution is put on the transition kernel. When one uses neural nets to estimate the model, SAMP can be approximated by epistemic neural networks [[Osband et al., 2021a](#)], a general framework to quantify uncertainty for neural nets. The effectiveness of different epistemic neural networks such as deep ensemble, dropout and stochastic gradient MCMC has been examined empirically by [Osband et al.](#) [[2021b]](#).*

We compute $\pi_{\text{r-IDS}}^\ell$ in two steps:

- Firstly, we prove an equivalent form of the objective function in Eq. (4.3) using the chain rule of mutual information. Define $r_h'(s, a)$ as an *augmented* reward function:

$$r_h'(s, a) = r_h(s, a) + \lambda \int D_{\mathrm{KL}}\left(P_h^\mathcal{E}(\cdot|s, a) || P_h^{\bar{\mathcal{E}}_\ell}(\cdot|s, a)\right) d\mathbb{P}_\ell(\mathcal{E}).$$

**Proposition 4.8.** *The following equivalence holds*

$$\mathbb{E}_\ell[V_{1,\pi}^\mathcal{E}(s_1^\ell)] + \lambda \mathbb{I}_\ell^\pi\left(\mathcal{E}; \mathcal{H}_{\ell,H}\right) = \mathbb{E}_\pi^{\bar{\mathcal{E}}_\ell}\left[\sum_{h=1}^H r_h'(s_h, a_h)\right].$$

The proof is deferred to Appendix A.4.

- Given *SAMP*, the augmented reward $r_h'$ and the MDP $\bar{\mathcal{E}}_\ell$ can be well approximated by Monte Carlo sampling. Therefore, at each episode $\ell$, finding $\pi_{\text{r-IDS}}^\ell$ is equivalent to find an optimal policy based on a computable and augmented MDP $\{P_h^{\bar{\mathcal{E}}_\ell}, r_h'\}_{h=1}^H$. This can be solved efficiently by any dynamic programming solver such as value iteration or policy iteration.

In the end, we show that $\pi_{\text{r-IDS}}^\ell$ enjoys the same regret bound as `vanilla-IDS` when the tunable parameter is carefully chosen.

**Theorem 4.9.** *By choosing* $\lambda = \sqrt{L\mathbb{E}[\Gamma^*]/\mathbb{I}(\mathcal{E}; \mathcal{D}_{L+1})}$, *we have*

$$\mathfrak{BR}_L(\pi^{\text{r-IDS}}) \leq \sqrt{\frac{3}{2}L\mathbb{E}[\Gamma^*]\mathbb{I}(\mathcal{E}; \mathcal{D}_{L+1})}\,.$$

The proof is deferred to Appendix A.5. Let $M_1, M_2$ be upper bounds of $\mathbb{E}[\Gamma^*]$ and $\mathbb{I}(\mathcal{E}; \mathcal{D}_{L+1})$ respectively. In practice, we could conservatively choose $\lambda = \sqrt{LM_1/M_2}$ such that $\mathfrak{BR}_L(\pi^{\text{r-IDS}}) \leq \sqrt{3/2M_1M_2L}$. From Lemmas 4.2 and 4.3 for tabular MDPs, we could choose $M_1 = 2SAH^3$ and $M_2 = 2S^2AH\log(SLH)$.

**Remark 4.10.** *Russo and Van Roy [2018, Section 9.3] also considered a tunable version of IDS (for bandits) but took a square form of* $\mathbb{E}_\ell[V_{1,\pi}^{\mathcal{E}}(s_1^\ell)]$. *While this makes no difference in bandits setting, this prevented us to use dynamic programming solver in RL setting. We are also inspired by Foster et al. [2021, Section 9.3] who studied the relationship between information ratio and Decision-Estimation Coefficient.*

# 5  Learning a surrogate environment

When the state space is large, the agent could take too much information to learn exactly the whole environment $\mathcal{E}$ which is reflected through $\mathbb{I}(\mathcal{E}; \mathcal{D}_{L+1})$. A key observation is that different states may correspond to the same value function who eventually determines the behavior of the optimal policy. Based on the rate-distortion theory developed in Dong and Van Roy [2018], we reduce this redundancy and construct a surrogate environment that needs less information to learn.

## 5.1  A rate distortion approach

The rate-distortion theory [Cover and Thomas, 1991] addresses the problem of determining the minimal number of bits per symbol that should be communicated over a channel, so that the source (input signal) can be approximately reconstructed at the receiver (output signal) without exceeding an expected distortion. It was recently introduced to bandits community to develop sharper bounds for linear bandits [Dong and Van Roy, 2018] and time-sensitive bandits [Russo and Van Roy, 2022]. We take a similar approach to construct a surrogate environment.

**Surrogate environment**   Suppose there exists a partition $\{\Theta_k\}_{k=1}^K$ over $\Theta$ such that for any $\mathcal{E}, \mathcal{E}' \in \Theta_k$ and any $k \in [K]$, we have

$$V_{1,\pi_{\mathcal{E}}^*}^{\mathcal{E}}(s_1^\ell) - V_{1,\pi_{\mathcal{E}}^*}^{\mathcal{E}'}(s_1^\ell) \leq \varepsilon\,, \tag{5.1}$$

where $\varepsilon > 0$ is the distortion tolerance and we write the optimal policy explicitly depending on the environment. Let $\zeta$ be a discrete random variable taking values in $\{1, \ldots, K\}$ that indicates the region $\mathcal{E}$ lies such that $\zeta = k$ if and only if $\mathcal{E} \in \Theta_k$. Therefore, $\zeta$ can be viewed as a statistic of $\mathcal{E}$ and less informative than $\mathcal{E}$ if $K$ is small.

The next lemma shows the existence of the surrogate environment based on the partition.

**Lemma 5.1.** *For any partition* $\{\Theta_k\}_{k=1}^K$ *and any* $\ell \in [L]$, *we can construct a surrogate environment* $\widetilde{\mathcal{E}}_\ell^* \in \Theta$ *which is a random MDP such that the law of* $\widetilde{\mathcal{E}}_\ell^*$ *only depends on* $\zeta$ *and*

$$\mathbb{E}_\ell\left[V_{1,\pi_{\mathcal{E}}^*}^{\mathcal{E}}(s_1^\ell) - V_{1,\pi_{TS}^\ell}^{\mathcal{E}}(s_1^\ell)\right] - \mathbb{E}_\ell\left[V_{1,\pi_{\mathcal{E}}^*}^{\widetilde{\mathcal{E}}_\ell^*}(s_1^\ell) - V_{1,\pi_{TS}^\ell}^{\widetilde{\mathcal{E}}_\ell^*}(s_1^\ell)\right] \leq \varepsilon\,. \tag{5.2}$$

The concrete form of $\widetilde{\mathcal{E}}_\ell^*$ is deferred to Eq. (B.1) in the appendix.

**Surrogate IDS**  We refer the IDS based on the surrogate environment $\widetilde{\mathcal{E}}_\ell^*$ as `surrogate-IDS` that minimizes

$$\pi_{\text{s-IDS}}^\ell = \underset{\pi \in \Pi}{\arg\min} \frac{(\mathbb{E}_\ell[V_{1,\pi^*}^{\mathcal{E}}(s_1^\ell) - V_{1,\pi}^{\mathcal{E}}(s_1^\ell)] - \varepsilon)^2}{\mathbb{I}_\ell^\pi(\widetilde{\mathcal{E}}_\ell^*; \mathcal{H}_{\ell,H})}, \tag{5.3}$$

for some parameters $\varepsilon > 0$ the will be chosen later. Denote the surrogate information ratio of TS as

$$\widetilde{\Gamma} = \max_{\ell \in [L]} \frac{\left(\mathbb{E}_\ell\left[V_{1,\pi^*}^{\widetilde{\mathcal{E}}_\ell^*}(s_1^\ell) - V_{1,\pi_{\text{TS}}^\ell}^{\widetilde{\mathcal{E}}_\ell^*}(s_1^\ell)\right]\right)^2}{\mathbb{I}_\ell^{\pi_{\text{TS}}^\ell}(\widetilde{\mathcal{E}}_\ell^*; \mathcal{H}_{\ell,H})}.$$

We first derive a generic regret bound for surrogate IDS in the following theorem.

**Theorem 5.2.** *A generic regret bound for surrogate IDS is*

$$\mathfrak{BR}_L(\pi_{\text{s-IDS}}) \leq \sqrt{\mathbb{E}[\widetilde{\Gamma}]\mathbb{I}(\zeta; \mathcal{D}_{L+1})L} + L\varepsilon.$$

We defer the proof to Appendix B.2. Given $\zeta$, we have $\widetilde{\mathcal{E}}_\ell^*$ and $\mathcal{H}_{\ell,H}$ are independent under the law of $\mathbb{P}_{\ell,\pi_{\text{s-IDS}}^\ell}$. By the data processing inequality, the proof uses the fact that

$$\mathbb{I}_\ell^{\pi_{\text{s-IDS}}^\ell}(\widetilde{\mathcal{E}}_\ell^*; \mathcal{H}_{\ell,H}) \leq \mathbb{I}_\ell^{\pi_{\text{s-IDS}}^\ell}(\zeta; \mathcal{H}_{\ell,H}).$$

Comparing with regret bound of `vanilla-IDS` in Lemma 4.1, the regret bound of `surrogate-IDS` depends on the information gain about $\zeta$ rather than the whole environment $\mathcal{E}$. If there exists a partition with small covering number $K$, the agent could pay less information to learn. The second term $L\varepsilon$ is the price of distortions.

In the following, we will bound the $\mathbb{E}[\widetilde{\Gamma}]$ and $\mathbb{I}(\zeta; \mathcal{D}_{L+1})$ for tabular and linear MDPs separately.

## 5.2  Tabular MDPs

We first show the existence of the partition required in Lemma 5.1 for tabular MDPs and an upper bound of the covering number $K$.

**Lemma 5.3.** *There exists a partition $\{\Theta_k^\varepsilon\}_{k=1}^K$ over $\Theta$ such that for any $k \in [K]$ and $\mathcal{E}_1, \mathcal{E}_2 \in \Theta_k^\varepsilon$,*

$$V_{1,\pi_{\mathcal{E}_1}^*}^{\mathcal{E}_1}(s_1) - V_{1,\pi_{\mathcal{E}_1}^*}^{\mathcal{E}_2}(s_1) \leq \varepsilon,$$

*and the log covering number satisfies $\log(K) \leq SAH \log(4H^2/\varepsilon)$.*

The proof is deferred to Lemma B.3. For tabular MDPs, the mutual information between $\zeta$ and the history can be bounded by

$$\mathbb{I}(\zeta; \mathcal{D}_{L+1}) \leq \mathbb{H}(\zeta) \leq \log(K) \leq SAH \log(4H^2/\varepsilon),$$

where $\mathbb{H}(\cdot)$ is the Shannon entropy. Comparing with Lemma 4.3 when learning the whole environment, learning the surrogate environment saves a factor of $S$ through the bound of mutual information.

**Lemma 5.4.** *The surrogate information ratio for tabular MDPs is upper bounded by*

$$\mathbb{E}[\widetilde{\Gamma}] \leq 2SAH^3.$$

The proof is the same as Lemma 4.2 and thus is omitted. Putting Lemmas 5.3-5.4 yields an improved bound for tabular MDPs using `surrogate-IDS`.

**Theorem 5.5** (Improved regret bound for tabular MDPs). *By choosing $\varepsilon = 1/L$, the regret bound of* `surrogate-IDS` *for tabular MDPs satisfies*

$$\mathfrak{BR}_L(\pi_{\text{s-IDS}}) \leq \sqrt{2S^2A^2H^4L \log(4HL)}.$$

For tabular MDPs, `surrogate-IDS` improves the regret bound of `vanilla-IDS` by a factor of $S$. However, it is still away from the minimax lower bound by a factor of $\sqrt{SAH}$. We conjecture `surrogate-IDS` can achieve the optimal bound with a price of lower order term but leave it as a future work.

**Remark 5.6.** *Although the existence of $\widetilde{\mathcal{E}}_\ell^*$ is established using a constructive argument, finding $\widetilde{\mathcal{E}}_\ell^*$ needs a grid search and is not computationally efficient.*

## 5.3 Linear MDPs

We extend our analysis to linear MDPs that is a fundamental model to study the theoretical properties of linear function approximations in RL. All the proofs are deferred to Appendix B.4-B.5.

**Definition 5.7** (Linear MDPs [Yang and Wang, 2019, Jin et al., 2020]). *Let $\phi : \mathcal{S} \times \mathcal{A} \to \mathbb{R}^d$ be a feature map which assigns to each state-action pair a $d$-dimensional feature vector and assume $\|\phi(s,a)\|_2 \leq 1$. An MDP is called a linear MDP if for any $h \in [H]$, there exist $d$ unknown (signed) measures $\psi_h^1, \ldots, \psi_h^d$ over $\mathcal{S}$, such that for any $(s,a) \in \mathcal{S} \times \mathcal{A}$, we have*

$$P_h(\cdot|s,a) = \langle \phi(s,a), \psi_h(\cdot) \rangle \,,$$

*where $\psi_h = (\psi_h^1, \ldots, \psi_h^d)$. Let us denote $\Theta^{Lin}$ be the parameter space of linear MDPs and assume $\|\sum_{s'} \psi_h(s')\|_2 \leq C_\psi$.*

Note that the degree of freedom of linear MDPs still depends on $S$ which implies that $\mathbb{I}(\mathcal{E}; \mathcal{D}_{L+1})$ may still scale with $S$. Therefore, we *must* learn a surrogate environment rather than the whole environment for linear MDPs based on the current regret decomposition in Theorem 4.4. We first show the existence of a partition over linear MDPs with the log covering number only depending on the feature dimension $d$.

**Lemma 5.8.** *There exists a partition $\{\Theta_k^\varepsilon\}_{k=1}^K$ over $\Theta^{Lin}$ such that for any $k \in [K]$ and $\mathcal{E}_1, \mathcal{E}_2 \in \Theta_k$,*

$$V_{1,\pi_{\mathcal{E}_1}^*}^{\mathcal{E}_1}(s_1) - V_{1,\pi_{\mathcal{E}_1}^*}^{\mathcal{E}_2}(s_1) \leq \varepsilon \,,$$

*and the log covering number satisfies $\log(K) \leq Hd \log(H^2 C_\psi/\varepsilon + 1)$.*

For linear MDPs, the mutual information can be bounded by

$$\mathbb{I}(\zeta; \mathcal{D}_{L+1}) \leq \mathbb{H}(\zeta) \leq \log(K) \leq Hd \log(H^2 C_\psi/\varepsilon + 1) \,.$$

**Lemma 5.9.** *The surrogate information ratio of linear MDPs is upper bounded by $\mathbb{E}[\widetilde{\Gamma}] \leq 4H^3 d$.*

**Theorem 5.10** (Regret bound for linear MDPs). *By choosing $\varepsilon = 1/L$, the regret bound of surrogate IDS for linear MDPs satisfies*

$$\mathfrak{BR}_L(\pi_{s\text{-}IDS}) \leq \sqrt{4H^4 d^2 L \log(H^2 C_\psi L + 1)} + 1 \,.$$

This Bayesian bound improves the $O(d^{3/2} H^2 \sqrt{L})$ frequentist regret of LSVI-UCB [Jin et al., 2020] by a factor of $\sqrt{d}$ and matches the existing minimax lower bound $O(\sqrt{H^3 d^2 L})$ [Zhou et al., 2021] up to a $H$ factor. However, we would like to emphasize that this is not an apples-to-apples comparison, since in general frequentist regret bound is stronger than Bayesian regret bound.

## 5.4 Regret bounds for TS

As a direct application of our rate-distortion analysis, we provide Bayesian regret bounds for Thompson sampling.

**Theorem 5.11.** *A generic regret bound for TS is*

$$\mathfrak{BR}_L(\pi_{TS}) \leq \sqrt{\mathbb{E}[\widetilde{\Gamma}] \mathbb{I}(\zeta; \mathcal{D}_{L+1}) L} + L\varepsilon \,.$$

This implies for tabular and linear MDPs, TS has the same regret bound as `surrogate-IDS`. Note that the computation of TS does not need to involve the surrogate environment $\widetilde{\mathcal{E}}_\ell^*$ so once the posterior sampling oracle is available, computing the policy is efficient. Howeverin when the worst-case information ratio cannot be optimally bounded by the information ratio of TS, IDS demonstrates better regret bounds than TS, such as bandits with graph feedback [Hao et al., 2022] and sparse linear bandits [Hao et al., 2021].

## 6 Conclusion

In this paper, we derive the first prior-free Bayesian regret bounds for information-directed RL under tabular and linear MDPs. Theoretically, it will be of great interest to see if any version of IDS can achieve the $O(\sqrt{SAH^3L})$ minimax lower bounds for tabular MDPs.

## Acknowledgements

We would like to thank Johannes Kirschner for helpful initial discussions.

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
