# A Proofs of learning the whole environment

## A.1 Proof of Theorem 4.1

*Proof.* We follow the standard information-theoretical regret decomposition:

$$
\mathfrak{BR}_L(\pi_{\text{IDS}}) = \sum_{\ell=1}^{L} \mathbb{E}\left[\mathbb{E}_\ell\left[V_{1,\pi^*}^{\mathcal{E}}(s_1^\ell) - V_{1,\pi_{\text{IDS}}^\ell}^{\mathcal{E}}(s_1^\ell)\right]\right]
$$

$$
= \sum_{\ell=1}^{L} \mathbb{E}\left[\sqrt{\frac{\left(\mathbb{E}_\ell\left[V_{1,\pi^*}^{\mathcal{E}}(s_1^\ell) - V_{1,\pi_{\text{IDS}}^\ell}^{\mathcal{E}}(s_1^\ell)\right]\right)^2}{\mathbb{I}_\ell^{\pi_{\text{IDS}}^\ell}(\mathcal{E};\mathcal{H}_{\ell,H})}\mathbb{I}_\ell^{\pi_{\text{IDS}}^\ell}(\mathcal{E};\mathcal{H}_{\ell,H})}\right]
$$

$$
\leq \sqrt{\mathbb{E}\left[\sum_{\ell=1}^{L}\Gamma_\ell(\pi_{\text{IDS}}^\ell)\right]}\sqrt{\mathbb{E}\left[\sum_{\ell=1}^{L}\mathbb{I}_\ell^{\pi_{\text{IDS}}^\ell}(\mathcal{E};\mathcal{H}_{\ell,H})\right]} = \sqrt{\mathbb{I}(\mathcal{E};\mathcal{D}_{L+1})\sum_{\ell=1}^{L}\mathbb{E}[\Gamma_\ell(\pi_{\text{IDS}}^\ell)]},
$$

where the first inequality is from the Cauchy–Schwarz inequality and the last equation is due to the chain rule of mutual information

$$
\mathbb{I}(\mathcal{E};\mathcal{D}_{L+1}) = \mathbb{I}(\mathcal{E};(\mathcal{H}_{1,H},\ldots,\mathcal{H}_{L,H})) = \sum_{\ell=1}^{L}\mathbb{E}\left[\mathbb{I}_\ell^{\pi_{\text{IDS}}^\ell}(\mathcal{E};\mathcal{H}_{\ell,H})\right].
$$

According to the definition of $\Gamma^*$, we have $\Gamma_\ell(\pi_{\text{IDS}}^\ell) \leq \Gamma^*$ for any $\ell \in [L]$. $\qquad\square$

## A.2 Proof of Lemma 4.2

*Proof.* From the definition of IDS policy stated in Eq. (4.2), for any $\ell \in [L]$,

$$
\Gamma_\ell(\pi_{\text{IDS}}^\ell) \leq \Gamma_\ell(\pi_{\text{TS}}^\ell) = \frac{\mathbb{E}_\ell\left[V_{1,\pi^*}^{\mathcal{E}}(s_1^\ell) - V_{1,\pi_{\text{TS}}^\ell}^{\mathcal{E}}(s_1^\ell)\right]^2}{\mathbb{I}_\ell^{\pi_{\text{TS}}^\ell}(\mathcal{E};\mathcal{H}_{\ell,H})}.
$$

We first decompose the one-step regret as follows,

$$
\mathbb{E}_\ell\left[V_{1,\pi^*}^{\mathcal{E}}(s_1^\ell) - V_{1,\pi_{\text{TS}}^\ell}^{\mathcal{E}}(s_1^\ell)\right] = \underbrace{\mathbb{E}_\ell\left[V_{1,\pi^*}^{\mathcal{E}}(s_1^\ell) - V_{1,\pi_{\text{TS}}^\ell}^{\bar{\mathcal{E}}_\ell}(s_1^\ell)\right]}_{I_1} + \underbrace{\mathbb{E}_\ell\left[V_{1,\pi_{\text{TS}}^\ell}^{\bar{\mathcal{E}}_\ell}(s_1^\ell) - V_{1,\pi_{\text{TS}}^\ell}^{\mathcal{E}}(s_1^\ell)\right]}_{I_2}.
$$

$$
\text{(A.1)}
$$

**Bounding $I_1$** Note that conditional on $\mathcal{D}_\ell$, the law of $\pi_{\text{TS}}^\ell$ is the same as the law of $\pi^*$ and both $\pi^*$ and $\pi_{\text{TS}}^\ell$ are independent of $\bar{\mathcal{E}}_\ell$. This fact implies that

$$
I_1 = \mathbb{E}_\ell\left[V_{1,\pi^*}^{\mathcal{E}}(s_1^\ell) - V_{1,\pi^*}^{\bar{\mathcal{E}}_\ell}(s_1^\ell)\right].
$$

Denote

$$
\Delta_h^{\mathcal{E}}(s,a) = \mathbb{E}_{s'\sim P_h^{\mathcal{E}}(\cdot|s,a)}[V_{h+1,\pi^*}^{\mathcal{E}}(s')] - \mathbb{E}_{s'\sim P_h^{\bar{\mathcal{E}}_\ell}(\cdot|s,a)}[V_{h+1,\pi^*}^{\mathcal{E}}(s')].
$$

Applying Lemma D.3, we have

$$
I_1 = \mathbb{E}_\ell\left[\sum_{h=1}^{H}\mathbb{E}_{\pi^*}^{\bar{\mathcal{E}}_\ell}\left[\Delta_h^{\mathcal{E}}(s_h^\ell,a_h^\ell)\right]\right] = \mathbb{E}_\ell\left[\sum_{h=1}^{H}\mathbb{E}\left[\mathbb{E}_{\pi^*}^{\bar{\mathcal{E}}_\ell}\left[\Delta_h^{\mathcal{E}}(s_h^\ell,a_h^\ell)\right]\big|\mathcal{E}\right]\right]
$$

$$
= \sum_{h=1}^{H}\mathbb{E}_\ell\left[\mathbb{E}\left[\sum_{(s,a)}d_{h,\pi^*}^{\bar{\mathcal{E}}_\ell}(s,a)\Delta_h^{\mathcal{E}}(s,a)\Big|\mathcal{E}\right]\right],
$$

$$
= \sum_{h=1}^{H}\mathbb{E}_\ell\left[\sum_{(s,a)}d_{h,\pi^*}^{\bar{\mathcal{E}}_\ell}(s,a)\Delta_h^{\mathcal{E}}(s,a)\right],
$$

$$
= \sum_{h=1}^{H}\mathbb{E}_\ell\left[\sum_{(s,a)}\frac{d_{h,\pi^*}^{\bar{\mathcal{E}}_\ell}(s,a)}{(\mathbb{E}_\ell[d_{h,\pi^*}^{\bar{\mathcal{E}}_\ell}(s,a)])^{1/2}}(\mathbb{E}_\ell[d_{h,\pi^*}^{\bar{\mathcal{E}}_\ell}(s,a)])^{1/2}\Delta_h^{\mathcal{E}}(s,a)\right].
$$

Here we just divide the element that corresponds to $d_{h,\pi^*}^{\bar{\mathcal{E}}_\ell}(s,a) > 0$.

Applying Cauchy–Schwarz inequality and using the fact that $d_{h,\pi^*}^{\bar{\mathcal{E}}_\ell}(s,a) \leq 1$, we have

$$
\begin{aligned}
I_1 &\leq \left( \sum_{h=1}^{H} \mathbb{E}_\ell \left[ \sum_{(s,a)} \frac{(d_{h,\pi^*}^{\bar{\mathcal{E}}_\ell}(s,a))^2}{\mathbb{E}_\ell[d_{h,\pi^*}^{\bar{\mathcal{E}}_\ell}(s,a)]} \right] \right)^{1/2} \left( \sum_{h=1}^{H} \mathbb{E}_\ell \left[ \sum_{(s,a)} \mathbb{E}_\ell[d_{h,\pi^*}^{\bar{\mathcal{E}}_\ell}(s,a)](\Delta_h^{\mathcal{E}}(s,a))^2 \right] \right)^{1/2} \\
&\leq \left( \sum_{h=1}^{H} \mathbb{E}_\ell \left[ \sum_{(s,a)} \frac{d_{h,\pi^*}^{\bar{\mathcal{E}}_\ell}(s,a)}{\mathbb{E}_\ell[d_{h,\pi^*}^{\bar{\mathcal{E}}_\ell}(s,a)]} \right] \right)^{1/2} \left( \sum_{h=1}^{H} \mathbb{E}_\ell \left[ \sum_{(s,a)} \mathbb{E}_\ell[d_{h,\pi^*}^{\bar{\mathcal{E}}_\ell}(s,a)](\Delta_h^{\mathcal{E}}(s,a))^2 \right] \right)^{1/2} .
\end{aligned}
\tag{A.2}
$$

First, note that

$$
\sum_{h=1}^{H} \mathbb{E}_\ell \left[ \sum_{(s,a)} \frac{d_{h,\pi^*}^{\bar{\mathcal{E}}_\ell}(s,a)}{\mathbb{E}_\ell[d_{h,\pi^*}^{\bar{\mathcal{E}}_\ell}(s,a)]} \right] = \sum_{h=1}^{H} \left[ \sum_{(s,a)} \frac{\mathbb{E}_\ell[d_{h,\pi^*}^{\bar{\mathcal{E}}_\ell}(s,a)]}{\mathbb{E}_\ell[d_{h,\pi^*}^{\bar{\mathcal{E}}_\ell}(s,a)]} \right] \leq HSA .
\tag{A.3}
$$

Second, since the law of $\pi^*$ is the same as $\pi_{\mathrm{TS}}^\ell$ conditional on $\mathcal{D}_\ell$, we have

$$
\mathbb{E}_\ell \left[ d_{h,\pi^*}^{\bar{\mathcal{E}}_\ell}(s,a) \right] = \mathbb{E}_\ell \left[ d_{h,\pi_{\mathrm{TS}}^\ell}^{\bar{\mathcal{E}}_\ell}(s,a) \right] .
$$

Then we can obtain

$$
\begin{aligned}
\mathbb{E}_\ell \left[ \sum_{(s,a)} \mathbb{E}_\ell[d_{h,\pi^*}^{\bar{\mathcal{E}}_\ell}(s,a)](\Delta_h^{\mathcal{E}}(s,a))^2 \right] &= \mathbb{E}_\ell \left[ \sum_{(s,a)} \mathbb{E}_\ell \left[ d_{h,\pi_{\mathrm{TS}}^\ell}^{\bar{\mathcal{E}}_\ell}(s,a) \right] (\Delta_h^{\mathcal{E}}(s,a))^2 \right] \\
&= \mathbb{E}_\ell \left[ \sum_{(s,a)} d_{h,\pi_{\mathrm{TS}}^\ell}^{\bar{\mathcal{E}}_\ell}(s,a) \right] \mathbb{E}_\ell \left[ (\Delta_h^{\mathcal{E}}(s,a))^2 \right] .
\end{aligned}
$$

Given $\mathcal{D}_\ell$, we have $d_{h,\pi_{\mathrm{TS}}^\ell}^{\bar{\mathcal{E}}_\ell}(s,a)$ and $\Delta_h^{\mathcal{E}}(s,a)$ are independent. This implies

$$
\begin{aligned}
\mathbb{E}_\ell &\left[ \sum_{(s,a)} d_{h,\pi_{\mathrm{TS}}^\ell}^{\bar{\mathcal{E}}_\ell}(s,a) \right] \mathbb{E}_\ell \left[ (\Delta_h^{\mathcal{E}}(s,a))^2 \right] = \mathbb{E}_\ell \left[ \sum_{(s,a)} d_{h,\pi_{\mathrm{TS}}^\ell}^{\bar{\mathcal{E}}_\ell}(s,a)(\Delta_h^{\mathcal{E}}(s,a))^2 \right] \\
&= \mathbb{E}_\ell \left[ \mathbb{E}_{\pi_{\mathrm{TS}}^\ell}^{\bar{\mathcal{E}}_\ell} \left[ (\Delta_h^{\mathcal{E}}(s_h^\ell, a_h^\ell))^2 \right] \right] \\
&= H^2 \mathbb{E}_\ell \left[ \mathbb{E}_{\pi_{\mathrm{TS}}^\ell}^{\bar{\mathcal{E}}_\ell} \left[ \left( \mathbb{E}_{s' \sim P_h^{\mathcal{E}}(\cdot|s_h^\ell, a_h^\ell)} \left[ V_{h+1,\pi^*}^{\mathcal{E}}(s')/H \right] - \mathbb{E}_{s' \sim P_h^{\bar{\mathcal{E}}_\ell}(\cdot|s_h^\ell, a_h^\ell)} \left[ V_{h+1,\pi^*}^{\mathcal{E}}(s')/H \right] \right)^2 \right] \right] ,
\end{aligned}
\tag{A.4}
$$

where $\mathbb{E}_{\pi_{\mathrm{TS}}^\ell}^{\bar{\mathcal{E}}_\ell}$ is taken with respect to $s_h^\ell, a_h^\ell$ and $\mathbb{E}_\ell$ is taken with respect to $\pi_{\mathrm{TS}}^\ell$ and $\mathcal{E}$.

Combining Eqs. (A.2)-(A.4) together and applying Lemma D.2 which is a variant of Pinsker's inequality,

$$
I_1 \leq \sqrt{SAH^3} \left( \sum_{h=1}^{H} \mathbb{E}_\ell \mathbb{E}_{\pi_{\mathrm{TS}}^\ell}^{\bar{\mathcal{E}}_\ell} \left[ \frac{1}{2} D_{\mathrm{KL}} \left( P_h^{\mathcal{E}}(\cdot|s_h^\ell, a_h^\ell) || P_h^{\bar{\mathcal{E}}_\ell}(\cdot|s_h^\ell, a_h^\ell) \right) \right] \right)^{1/2} .
\tag{A.5}
$$

The next lemma establishes the relationship between the cumulative KL-distance in Eq. (A.5) and mutual information $\mathbb{I}_\ell^{\pi_{\mathrm{TS}}^\ell}(\mathcal{E}; \mathcal{H}_{\ell,H})$. This lemma will be frequently used later.

**Lemma A.1.** *For environment $\mathcal{E}$ and its corresponding mean of posterior measure $\bar{\mathcal{E}}_\ell$, the following holds for any policy $\pi$,*

$$
\sum_{h=1}^{H} \mathbb{E}_\ell \mathbb{E}_\pi^{\bar{\mathcal{E}}_\ell} \left[ D_{\mathrm{KL}} \left( P_h^{\mathcal{E}}(\cdot|s_h^\ell, a_h^\ell) || P_h^{\bar{\mathcal{E}}_\ell}(\cdot|s_h^\ell, a_h^\ell) \right) \right] = \mathbb{I}_\ell^\pi (\mathcal{E}; \mathcal{H}_{\ell,H}) .
$$

Together with Eq. (A.5),

$$I_1 \leq \sqrt{\frac{1}{2} SAH^3 \mathbb{I}_\ell^{\pi_{\mathrm{TS}}^\ell} (\mathcal{E}; \mathcal{H}_{\ell,H})} . \tag{A.6}$$

**Bounding $I_2$**  The upper bound for $I_2$ is similar. Applying Lemma D.3 and repeating the steps of showing Eq. (A.5),

$$\mathbb{E}_\ell \left[ V_{1,\pi_{\mathrm{TS}}^\ell}^{\mathcal{E}}(s_1^\ell) - V_{1,\pi_{\mathrm{TS}}^\ell}^{\bar{\mathcal{E}}_\ell}(s_1^\ell) \right]$$

$$= \mathbb{E}_\ell \left[ \sum_{h=1}^{H} \mathbb{E}_{\pi_{\mathrm{TS}}^\ell}^{\bar{\mathcal{E}}_\ell} \left[ \mathbb{E}_{s' \sim P_h^{\mathcal{E}}(\cdot|s_h^\ell, a_h^\ell)}[V_{h+1,\pi_{\mathrm{TS}}^\ell}^{\mathcal{E}}(s')] - \mathbb{E}_{s' \sim P_h^{\bar{\mathcal{E}}_\ell}(\cdot|s_h^\ell, a_h^\ell)}[V_{h+1,\pi_{\mathrm{TS}}^\ell}^{\mathcal{E}}(s')] \right] \right]$$

$$\leq \sqrt{SAH^3} \left( \sum_{h=1}^{H} \mathbb{E}_\ell \mathbb{E}_{\pi_{\mathrm{TS}}^\ell}^{\bar{\mathcal{E}}_\ell} \left[ \frac{1}{2} D_{\mathrm{KL}} \left( P_h^{\mathcal{E}}(\cdot|s_h^\ell, a_h^\ell) || P_h^{\bar{\mathcal{E}}_\ell}(\cdot|s_h, a_h) \right) \right] \right)^{1/2} .$$

Applying Lemma A.1 again, we obtain

$$I_2 \leq \sqrt{SAH^3 \mathbb{I}_\ell^{\pi_{\mathrm{TS}}^\ell} (\mathcal{E}; \mathcal{H}_{\ell,H})} \tag{A.7}$$

Combining Eqs. (A.1), (A.6) and (A.7) together, we have

$$\Gamma^* \leq \max_{\ell \in [L]} \Gamma_\ell(\pi_{\mathrm{TS}}^\ell) = \frac{\left( \mathbb{E}_\ell \left[ V_{1,\pi_{\mathrm{TS}}^\ell}^{\mathcal{E}}(s_1^\ell) - V_{1,\pi^*}^{\mathcal{E}}(s_1^\ell) \right] \right)^2}{\mathbb{I}_\ell^{\pi_{\mathrm{TS}}^\ell} (\mathcal{E}; \mathcal{H}_{\ell,H})} \leq 2SAH^3 .$$

This ends the proof. □

## A.3  Proof of Lemma 4.3

*Proof.* Now we start to bound $\mathbb{I}(\mathcal{E}; \mathcal{D}_{L+1})$. Assume the prior probability measure of $\mathcal{E}$ is $\rho(\mathcal{E})$ that takes the value in $\Theta$ and suppose the Bayes mixture density $p_\rho(\mathcal{D}_{L+1}) = \int_{\mathcal{E} \in \Theta} p(\mathcal{D}_{L+1}|\mathcal{E}) d\rho(\mathcal{E})$. Let $\{\Theta_k\}_{k=1}^{K}$ be a partition of $\Theta$. We choose $\rho_1$ as an uniform distribution over $\{\Theta_k\}_{k=1}^{K}$ such that $q(\mathcal{D}_{L+1}) = p_{\rho_1}(\mathcal{D}_{L+1}) = \int_{\theta \in \Theta} p(\mathcal{D}_{L+1}|\theta) \, d\rho_1(\theta)$ and we denote $\mathbb{Q}_{\mathcal{D}_{L+1}}$ as the corresponding probability measure. Based on the equivalent form of mutual information,

$$\mathbb{I}(\mathcal{E}; \mathcal{D}_{L+1}) = \mathbb{E}_{\mathcal{E}} \left[ D_{\mathrm{KL}}(\mathbb{P}_{\mathcal{D}_{L+1}|\mathcal{E}} || \mathbb{P}_{\mathcal{D}_{L+1}}) \right]$$

$$= \int_{\mathcal{E} \in \Theta} \int p(\mathcal{D}_{L+1}|\mathcal{E}) \log \left( \frac{p(\mathcal{D}_{L+1}|\mathcal{E})}{p_\rho(\mathcal{D}_{L+1})} \right) \mu(\mathrm{d}\mathcal{D}_{L+1}) \, \mathrm{d}\rho(\mathcal{E})$$

$$\leq \int_{\mathcal{E} \in \Theta} \int p(\mathcal{D}_{L+1}|\mathcal{E}) \log \left( \frac{p(\mathcal{D}_{L+1}|\mathcal{E})}{q(\mathcal{D}_{L+1})} \right) \mu(\mathrm{d}\mathcal{D}_{L+1}) \, \mathrm{d}\rho(\mathcal{E}) \tag{A.8}$$

$$= \int_{\mathcal{E} \in \Theta} D_{\mathrm{KL}}(\mathbb{P}_{\mathcal{D}_{L+1}|\mathcal{E}} || \mathbb{Q}_{\mathcal{D}_{L+1}}) \, \mathrm{d}\rho(\mathcal{E}) .$$

where the inequality is due to the fact that Bayes mixture density $p_\rho(\mathcal{D}_{L+1})$ minimizes the average KL divergences over any choice of densities $q(\mathcal{D}_{L+1})$.

According to the definition of the KL-divergence term,

$$D_{\mathrm{KL}} \left( \mathbb{P}_{\mathcal{D}_{L+1}|\mathcal{E}} || \mathbb{Q}_{\mathcal{D}_{L+1}} \right) = \mathbb{E} \left[ \log \frac{p(\mathcal{D}_{L+1}|\mathcal{E})}{1/K \sum_{\widetilde{\mathcal{E}} \in \Theta_k} p(\mathcal{D}_{L+1}|\widetilde{\mathcal{E}})} \right]$$

$$\leq \mathbb{E} \left[ \log \frac{p(\mathcal{D}_{L+1}|\mathcal{E})}{1/K p(\mathcal{D}_{L+1}|\widetilde{\mathcal{E}})} \right] \tag{A.9}$$

$$\leq \log(K) + D_{\mathrm{KL}} \left( \mathbb{P}_{\mathcal{D}_{L+1}|\mathcal{E}} || \mathbb{P}_{\mathcal{D}_{L+1}|\widetilde{\mathcal{E}}} \right) .$$

By the chain rule of KL-divergence,

$$D_{\mathrm{KL}}\left(\mathbb{P}_{\mathcal{D}_{L+1}|\mathcal{E}}||\mathbb{P}_{\mathcal{D}_{L+1}|\widetilde{\mathcal{E}}}\right) = \mathbb{E}\left[\sum_{\ell=1}^{L}\sum_{h=1}^{H}D_{\mathrm{KL}}\left(P(\cdot|s_h^\ell,a_h^\ell,\mathcal{E})||P(\cdot|s_h^\ell,a_h^\ell,\widetilde{\mathcal{E}})\right)\right]$$

$$= \mathbb{E}\left[\sum_{\ell=1}^{L}\sum_{h=1}^{H}D_{\mathrm{KL}}\left(P_h^{\mathcal{E}}(\cdot|s_h^\ell,a_h^\ell)||P_h^{\widetilde{\mathcal{E}}}(\cdot|s_h^\ell,a_h^\ell)\right)\right].$$

It remains to construct a partition of $\{\Theta_k\}_{k=1}^K$ such that for any $\mathcal{E} \in \Theta$, there always exists $\widetilde{\mathcal{E}} \in \Theta_k$ such that for any $s, a, h$,

$$D_{\mathrm{KL}}\left(P_h^{\mathcal{E}}(\cdot|s,a)||P_h^{\widetilde{\mathcal{E}}}(\cdot|s,a)\right) \le \varepsilon.$$

The existence of such partition is shown in the following theorem.

**Theorem A.2** (Divergence covering number). *For any $0 < \varepsilon < 1$, suppose $\mathcal{N}(k,\varepsilon)$ is a divergence covering set over $k$-dimensional probability simplex such that for any $P$, there exists $\widetilde{P} \in \mathcal{N}(k,\varepsilon)$ such that*

$$D_{\mathrm{KL}}(P||\widetilde{P}) \le \varepsilon.$$

*Then there exists such set whose covering number can be bounded by*

$$|\mathcal{N}(k,\varepsilon)| \le 8^{k-1}\left(\frac{k-1}{\varepsilon}\right)^{\frac{k-1}{2}}.$$

The proof could be found in Tang and Polyanskiy [2021, Theorem 4]. For each $s, a, h$, we construct such divergence covering set $\mathcal{N}_{sa}^h(S,\varepsilon)$ and the partition $\{\Theta_k\}_{k=1}^K$ is constructed such that $\mathcal{E} \in \Theta_k$ if for any $s, a, h$, $P_h^{\mathcal{E}}(\cdot|s,a) \in \mathcal{N}_{sa}^h(S,\varepsilon)$. According to Theorem A.2,

$$\log(K) \le \log\left(|\mathcal{N}_{sa}^h(S,\varepsilon)|^{SAH}\right) \le (S-1)SAH\log(8) + (S-1)SAH\log\left(\frac{S-1}{\varepsilon}\right).$$

Therefore, together with Eq. (A.9),

$$\mathbb{I}(\mathcal{E};\mathcal{D}_{L+1}) \le LH\varepsilon + S^2AH\log\left(S/\varepsilon\right).$$

By choosing $\varepsilon = 1/LH$,

$$\mathbb{I}(\mathcal{E};\mathcal{D}_{L+1}) \le 2S^2AH\log\left(SLH\right).$$

This ends the proof. $\qquad\square$

## A.4 Proof of Proposition 4.8

According to the linearity of expectation and independence of priors over different layers, we can obtain $\mathbb{E}_\ell[V_{1,\pi}^{\mathcal{E}}(s_1^\ell)] = V_{1,\pi}^{\bar{\mathcal{E}}_\ell}(s_1^\ell)$ which implies

$$\mathbb{E}_\ell[V_{1,\pi}^{\mathcal{E}}(s_1^\ell)] = \mathbb{E}_\pi^{\bar{\mathcal{E}}_\ell}\left[\sum_{h=1}^{H}r_h(s_h,a_h)\right].$$

According to the proof of Lemma A.1 in Appendix C.1

$$\mathbb{I}_\ell^\pi\left(\mathcal{E};\mathcal{H}_{\ell,H}\right) = \sum_{h=1}^{H}\mathbb{E}_\pi^{\bar{\mathcal{E}}_\ell}\left[\int D_{\mathrm{KL}}\left(P_h^{\mathcal{E}}(\cdot|s_h,a_h)||P_h^{\bar{\mathcal{E}}_\ell}(\cdot|s_h,a_h)\right)\mathrm{d}\mathbb{P}_\ell(\mathcal{E})\right].$$

Combining them together,

$$\mathbb{E}_\ell[V_{1,\pi}^{\mathcal{E}}(s_1^\ell)] + \lambda\mathbb{I}_\ell^\pi\left(\mathcal{E};\mathcal{H}_{\ell,H}\right)$$

$$=\mathbb{E}_\pi^{\bar{\mathcal{E}}_\ell}\left[\sum_{h=1}^{H}\left(r_h(s_h,a_h) + \lambda\int D_{\mathrm{KL}}\left(P_h^{\mathcal{E}}(\cdot|s_h,a_h)||P_h^{\bar{\mathcal{E}}_\ell}(\cdot|s_h,a_h)\right)\mathrm{d}\mathbb{P}_\ell(\mathcal{E})\right)\right].$$

## A.5 Proof of Theorem 4.9

*Proof.* Using the fact that $2ab \le a^2 + b^2$, we have for any policy $\pi$

$$\frac{\mathbb{E}_\ell \left[ V_{1,\pi^*}^{\mathcal{E}}(s_1^\ell) - V_{1,\pi}^{\mathcal{E}}(s_1^\ell) \right] \sqrt{\lambda \mathbb{I}_\ell^\pi (\mathcal{E}; \mathcal{H}_{\ell,H})}}{\sqrt{\lambda \mathbb{I}_\ell^\pi (\mathcal{E}; \mathcal{H}_{\ell,H})}} \le \frac{\left( \mathbb{E}_\ell \left[ V_{1,\pi^*}^{\mathcal{E}}(s_1^\ell) - V_{1,\pi}^{\mathcal{E}}(s_1^\ell) \right] \right)^2}{2\lambda \mathbb{I}_\ell^\pi (\mathcal{E}; \mathcal{H}_{\ell,H})} + \frac{\lambda}{2} \mathbb{I}_\ell^\pi (\mathcal{E}; \mathcal{H}_{\ell,H}) \,.$$

This implies

$$\mathbb{E}_\ell \left[ V_{1,\pi^*}^{\mathcal{E}}(s_1^\ell) - V_{1,\pi}^{\mathcal{E}}(s_1^\ell) \right] - \frac{\lambda}{2} \mathbb{I}_\ell^\pi (\mathcal{E}; \mathcal{H}_{\ell,H}) \le \frac{\left( \mathbb{E}_\ell \left[ V_{1,\pi^*}^{\mathcal{E}}(s_1^\ell) - V_{1,\pi}^{\mathcal{E}}(s_1^\ell) \right] \right)^2}{2\lambda \mathbb{I}_\ell^\pi (\mathcal{E}; \mathcal{H}_{\ell,H})} \,. \tag{A.10}$$

We follow the regret decomposition as

$$\mathfrak{BR}_L(\pi_{\text{r-IDS}})$$

$$= \sum_{\ell=1}^{L} \mathbb{E} \left[ \mathbb{E}_\ell \left[ V_{1,\pi^*}^{\mathcal{E}}(s_1^\ell) - V_{1,\pi_{\text{r-IDS}}^\ell}^{\mathcal{E}}(s_1^\ell) \right] \right]$$

$$= \mathbb{E} \left[ \sum_{\ell=1}^{L} \mathbb{E}_\ell \left[ V_{1,\pi^*}^{\mathcal{E}}(s_1^\ell) - V_{1,\pi_{\text{r-IDS}}^\ell}^{\mathcal{E}}(s_1^\ell) \right] - \lambda \sum_{\ell=1}^{L} \mathbb{I}_\ell^{\text{r-IDS}} (\mathcal{E}; \mathcal{H}_{\ell,H}) \right] + \mathbb{E} \left[ \lambda \sum_{\ell=1}^{L} \mathbb{I}_\ell^{\text{r-IDS}} (\mathcal{E}; \mathcal{H}_{\ell,H}) \right] \,.$$

From the definition of $\pi_{\text{r-IDS}}^\ell$ and Eq. (A.10), we have

$$\mathbb{E}_\ell \left[ V_{1,\pi^*}^{\mathcal{E}}(s_1^\ell) - V_{1,\pi_{\text{r-IDS}}^\ell}^{\mathcal{E}}(s_1^\ell) \right] - \frac{\lambda}{2} \mathbb{I}_\ell^{\pi_{\text{r-IDS}}^\ell} (\mathcal{E}; \mathcal{H}_{\ell,H}) \le \mathbb{E}_\ell \left[ V_{1,\pi^*}^{\mathcal{E}}(s_1^\ell) - V_{1,\pi_{\text{IDS}}^\ell}^{\mathcal{E}}(s_1^\ell) \right] - \frac{\lambda}{2} \mathbb{I}_\ell^{\pi_{\text{IDS}}^\ell} (\mathcal{E}; \mathcal{H}_{\ell,H})$$

$$\le \frac{\left( \mathbb{E}_\ell \left[ V_{1,\pi^*}^{\mathcal{E}}(s_1^\ell) - V_{1,\pi_{\text{IDS}}^\ell}^{\mathcal{E}}(s_1^\ell) \right] \right)^2}{2\lambda \mathbb{I}_\ell^{\pi_{\text{IDS}}^\ell} (\mathcal{E}; \mathcal{H}_{\ell,H})} \le \frac{\Gamma^*}{2\lambda} \,,$$

where $\Gamma^*$ is the worst-case information ratio. Overall, this implies

$$\mathfrak{BR}_L(\pi_{\text{r-IDS}}) \le \frac{L\mathbb{E}[\Gamma^*]}{2\lambda} + \lambda \mathbb{I} (\mathcal{E}; \mathcal{D}_{L+1}) \,.$$

Letting $\lambda = \sqrt{L\mathbb{E}[\Gamma^*]/\mathbb{I}(\mathcal{E}; \mathcal{D}_{L+1})}$, we have

$$\mathfrak{BR}_L(\pi^{\text{r-IDS}}) \le \sqrt{\frac{3}{2} L\mathbb{E}[\Gamma^*]\mathbb{I}(\mathcal{E}; \mathcal{D}_{L+1})} \,.$$

This ends the proof. $\qquad\square$

# B   Proofs of learning a surrogate environment

## B.1   Proof of Lemma 5.1

*Proof.* Conditional on $\mathcal{D}_\ell$, let $\mathcal{E}_\ell$ be an independent sample of $\mathcal{E}$. Note that

$$\mathbb{E}_\ell \left[ V_{1,\pi_{\mathcal{E}}^*}^{\mathcal{E}_\ell}(s_1^\ell) \big| \mathcal{E}_\ell \in \Theta_k \right] = \sum_{\mathcal{E} \in \Theta_k^\varepsilon} \mathbb{P} (\mathcal{E}_\ell = \mathcal{E} | \mathcal{E}_\ell \in \Theta_k) \, \mathbb{E}_\ell \left[ V_{1,\pi_{\mathcal{E}}^*}^{\mathcal{E}}(s_1^\ell) \big| \mathcal{E}_\ell \in \Theta_k \right]$$

$$= \sum_{\mathcal{E} \in \Theta_k^\varepsilon} \mathbb{P} (\mathcal{E}_\ell = \mathcal{E} | \mathcal{E}_\ell \in \Theta_k) \, \mathbb{E}_\ell \left[ V_{1,\pi_{\mathcal{E}}^*}^{\mathcal{E}}(s_1^\ell) \right] \,,$$

where the last equation is due to the independence between $\mathcal{E}_\ell$ and $\mathcal{E}$. For each $k \in [K]$, according to Lemma D.1, there exists $\mathcal{E}_1^{k,\ell}, \mathcal{E}_2^{k,\ell} \in \Theta_k^\varepsilon$ and $r_{k,\ell} \in [0,1]$ such that

$$r_{k,\ell} \mathbb{E}_\ell \left[ V_{1,\pi_{\mathcal{E}}^*}^{\mathcal{E}_1^{k,\ell}}(s_1^\ell) \right] + (1 - r_{k,\ell}) \mathbb{E}_\ell \left[ V_{1,\pi_{\mathcal{E}}^*}^{\mathcal{E}_2^{k,\ell}}(s_1^\ell) \right] \le \sum_{\mathcal{E} \in \Theta_k^\varepsilon} \mathbb{P} (\mathcal{E}_\ell = \mathcal{E} | \mathcal{E}_\ell \in \Theta_k) \, \mathbb{E}_\ell \left[ V_{1,\pi_{\mathcal{E}}^*}^{\mathcal{E}}(s_1^\ell) \right]$$

$$= \mathbb{E}_\ell \left[ V_{1,\pi_{\mathcal{E}}^*}^{\mathcal{E}_\ell}(s_1^\ell) \big| \mathcal{E}_\ell \in \Theta_k^\varepsilon \right] \,.$$

The surrogate learning target $\widetilde{\mathcal{E}}_\ell^*$ is a random variable such that

$$\mathbb{P}_\ell\left(\widetilde{\mathcal{E}}_\ell^* = \mathcal{E}_1^{k,\ell}\big|\zeta = k\right) = r_{k,\ell}, \mathbb{P}_\ell\left(\widetilde{\mathcal{E}}_\ell^* = \mathcal{E}_2^{k,\ell}\big|\zeta = k\right) = 1 - r_{k,\ell}. \tag{B.1}$$

This implies the law of $\widetilde{\mathcal{E}}_\ell^*$ only depends on $\zeta$ and conditional on $\zeta$, $\widetilde{\mathcal{E}}_\ell^*$ is independent of $\mathcal{E}$.

We also need to ensure learning towards $\widetilde{\mathcal{E}}_\ell^*$ will not occur too much additional regret. Let $\widetilde{\mathcal{E}}_\ell$ be an independent sample of $\widetilde{\mathcal{E}}_\ell^*$. From the law of total expectations,

$$\mathbb{E}_\ell\left[V_{1,\pi_{\mathcal{E}}^*}^{\widetilde{\mathcal{E}}_\ell}(s_1^\ell) - V_{1,\pi_{\mathcal{E}}^*}^{\mathcal{E}_\ell}(s_1^\ell)\right]$$

$$= \sum_{k=1}^K \mathbb{P}\left(\mathcal{E}_\ell \in \Theta_k^\varepsilon\right)\mathbb{E}_\ell\left[V_{1,\pi_{\mathcal{E}}^*}^{\widetilde{\mathcal{E}}_\ell}(s_1^\ell) - V_{1,\pi_{\mathcal{E}}^*}^{\mathcal{E}_\ell}(s_1^\ell)\big|\mathcal{E}_\ell \in \Theta_k^\varepsilon\right]$$

$$= \sum_{k=1}^K \mathbb{P}\left(\mathcal{E}_\ell \in \Theta_k^\varepsilon\right)\left(r_{k,\ell}\mathbb{E}_\ell\left[V_{1,\pi_{\mathcal{E}}^*}^{\mathcal{E}_1^{k,\ell}}(s_1^\ell)\right] + (1 - r_{k,\ell})\mathbb{E}_\ell\left[V_{1,\pi_{\mathcal{E}}^*}^{\mathcal{E}_2^{k,\ell}}(s_1^\ell)\right] - \mathbb{E}_\ell\left[V_{1,\pi_{\mathcal{E}}^*}^{\mathcal{E}_\ell}(s_1^\ell)\big|\mathcal{E}_\ell \in \Theta_k^\varepsilon\right]\right)$$

$$\leq 0.$$

On the other hand, we have $\mathbb{E}_\ell[V_{1,\pi_{\mathcal{E}}^*}^{\mathcal{E}_\ell}(s_1^\ell)] = \mathbb{E}_\ell[V_{1,\pi_{\text{TS}}^\ell}^{\mathcal{E}}(s_1^\ell)]$ and this implies

$$\mathbb{E}_\ell\left[V_{1,\pi_{\mathcal{E}}^*}^{\widetilde{\mathcal{E}}_\ell}(s_1^\ell) - V_{1,\pi_{\text{TS}}^\ell}^{\mathcal{E}}(s_1^\ell)\right] \leq 0. \tag{B.2}$$

When $\mathcal{E} \in \Theta_k^\varepsilon$, then $\zeta = k$ which implies $\widetilde{\mathcal{E}}_\ell^* \in \Theta_k^\varepsilon$ either. That means $\mathcal{E}$ and $\widetilde{\mathcal{E}}^*$ are in the same partition and

$$\mathbb{E}_\ell\left[V_{1,\pi_{\mathcal{E}}^*}^{\mathcal{E}}(s_1^\ell) - V_{1,\pi_{\mathcal{E}}^*}^{\widetilde{\mathcal{E}}_\ell^*}(s_1^\ell)\right] \leq \varepsilon. \tag{B.3}$$

Putting Eqs. (B.2)-(B.3) together,

$$\mathbb{E}_\ell\left[V_{1,\pi_{\mathcal{E}}^*}^{\mathcal{E}}(s_1^\ell) - V_{1,\pi_{\text{TS}}^\ell}^{\mathcal{E}}(s_1^\ell)\right] - \mathbb{E}_\ell\left[V_{1,\pi_{\mathcal{E}}^*}^{\widetilde{\mathcal{E}}_\ell^*}(s_1^\ell) - V_{1,\pi_{\mathcal{E}}^*}^{\widetilde{\mathcal{E}}_\ell}(s_1^\ell)\right] \leq \varepsilon. \tag{B.4}$$

Noticing that $\mathbb{E}_\ell[V_{1,\pi_{\mathcal{E}}^*}^{\widetilde{\mathcal{E}}_\ell}(s_1^\ell)] = \mathbb{E}_\ell[V_{1,\pi_{\text{TS}}^\ell}^{\widetilde{\mathcal{E}}_\ell^*}(s_1^\ell)]$, this ends the proof. $\square$

### B.2 Proof of Theorem 5.2

*Proof.* We decompose

$$\mathfrak{BR}_L(\pi_{\text{s-IDS}}) = \sum_{\ell=1}^L \mathbb{E}\left[\mathbb{E}_\ell\left[V_{1,\pi^*}^{\mathcal{E}}(s_1^\ell) - V_{1,\pi_{\text{s-IDS}}^\ell}^{\mathcal{E}}(s_1^\ell)\right] - \varepsilon\right] + L\varepsilon$$

$$\leq \sqrt{\mathbb{E}\left[\sum_{\ell=1}^L \frac{\left(\mathbb{E}_\ell\left[V_{1,\pi^*}^{\mathcal{E}}(s_1^\ell) - V_{1,\pi_{\text{s-IDS}}^\ell}^{\mathcal{E}}(s_1^\ell)\right] - \varepsilon\right)^2}{\mathbb{I}_\ell^{\pi_{\text{s-IDS}}^\ell}(\widetilde{\mathcal{E}}_\ell^*; \mathcal{H}_{\ell,H})}\right]}\sqrt{\mathbb{E}\left[\sum_{\ell=1}^L \mathbb{I}_\ell^{\pi_{\text{s-IDS}}^\ell}(\widetilde{\mathcal{E}}_\ell^*; \mathcal{H}_{\ell,H})\right]} + L\varepsilon.$$

From the definition of $\pi_{\text{s-IDS}}^\ell$ in Eq. (5.3),

$$\mathfrak{BR}_L(\pi_{\text{s-IDS}}) \leq \sqrt{\mathbb{E}\left[\sum_{\ell=1}^L \frac{\left(\mathbb{E}_\ell\left[V_{1,\pi^*}^{\mathcal{E}}(s_1^\ell) - V_{1,\pi_{\text{TS}}^\ell}^{\mathcal{E}}(s_1^\ell)\right] - \varepsilon\right)^2}{\mathbb{I}_\ell^{\pi_{\text{TS}}^\ell}(\widetilde{\mathcal{E}}_\ell^*; \mathcal{H}_{\ell,H})}\right]}\sqrt{\mathbb{E}\left[\sum_{\ell=1}^L \mathbb{I}_\ell^{\pi_{\text{r-IDS}}^\ell}(\widetilde{\mathcal{E}}_\ell^*; \mathcal{H}_{\ell,H})\right]} + L\varepsilon.$$

According to Lemma 5.1,

$$\mathbb{E}_\ell\left[V_{1,\pi^*}^{\mathcal{E}}(s_1^\ell) - V_{1,\pi_{\text{TS}}^\ell}^{\mathcal{E}}(s_1^\ell)\right] - \varepsilon \leq \mathbb{E}_\ell\left[V_{1,\pi^*}^{\widetilde{\mathcal{E}}_\ell^*}(s_1^\ell) - V_{1,\pi_{\text{TS}}^\ell}^{\widetilde{\mathcal{E}}_\ell^*}(s_1^\ell)\right].$$

For any $\ell \in [L]$, conditional on $\zeta$, we have $\widetilde{\mathcal{E}}_\ell^*$ and $\mathcal{H}_{\ell,H}$ are independent under the law of $\mathbb{P}_{\ell,\pi_{\text{s-IDS}}^\ell}$. By the data processing inequality, we have

$$\mathbb{I}_\ell^{\pi_{\text{s-IDS}}^\ell}(\widetilde{\mathcal{E}}_\ell^*; \mathcal{H}_{\ell,H}) \leq \mathbb{I}_\ell^{\pi_{\text{s-IDS}}^\ell}(\zeta; \mathcal{H}_{\ell,H}).$$

Therefore,

$$\mathfrak{BR}_L(\pi_{\text{s-IDS}}) \leq \sqrt{\mathbb{E}\left[\sum_{\ell=1}^{L} \frac{\left(\mathbb{E}_\ell\left[V_{1,\pi^*}^{\widetilde{\mathcal{E}}_\ell^*}(s_1^\ell) - V_{1,\pi_{\text{TS}}^\ell}^{\widetilde{\mathcal{E}}_\ell^*}(s_1^\ell)\right]\right)^2}{\mathbb{I}_\ell^{\pi_{\text{TS}}^\ell}(\widetilde{\mathcal{E}}_\ell^*; \mathcal{H}_{\ell,H})}\right]} \sqrt{\mathbb{E}\left[\sum_{\ell=1}^{L} \mathbb{I}_\ell^{\pi_{\text{s-IDS}}^\ell}(\zeta; \mathcal{H}_{\ell,H})\right]} + L\varepsilon$$

$$= \sqrt{\mathbb{E}\left[\sum_{\ell=1}^{L} \frac{\left(\mathbb{E}_\ell\left[V_{1,\pi^*}^{\widetilde{\mathcal{E}}_\ell^*}(s_1^\ell) - V_{1,\pi_{\text{TS}}^\ell}^{\widetilde{\mathcal{E}}_\ell^*}(s_1^\ell)\right]\right)^2}{\mathbb{I}_\ell^{\pi_{\text{TS}}^\ell}(\widetilde{\mathcal{E}}_\ell^*; \mathcal{H}_{\ell,H})}\right]} \sqrt{\mathbb{I}(\zeta; \mathcal{D}_{L+1})} + L\varepsilon.$$

This ends the proof. $\qquad\square$

### B.3 Proof of Lemma 5.3

*Proof.* We construct a partition over $\Theta$ such that Eq. (5.1) holds. For any $\mathcal{E}_1, \mathcal{E}_2 \in \Theta_k$, we use Lemma D.3,

$$V_{1,\pi_{\mathcal{E}_1}^*}^{\mathcal{E}_1}(s_1) - V_{1,\pi_{\mathcal{E}_1}^*}^{\mathcal{E}_2}(s_1)$$

$$= \sum_{h=1}^{H} \mathbb{E}_{\pi_{\mathcal{E}_1}^*}^{\mathcal{E}_2}\left[\mathbb{E}_{s'\sim P_h^{\mathcal{E}_1}(\cdot|s_h^\ell,a_h^\ell)}[V_{h+1,\pi_{\mathcal{E}_1}^*}^{\mathcal{E}_1}(s')] - \mathbb{E}_{s'\sim P_h^{\mathcal{E}_2}(\cdot|s_h^\ell,a_h^\ell)}[V_{h+1,\pi_{\mathcal{E}_1}^*}^{\mathcal{E}_1}(s')]\right]$$

$$= \sum_{h=1}^{H} \mathbb{E}_{\pi_{\mathcal{E}_1}^*}^{\mathcal{E}_2}\left[P_h^{\mathcal{E}_1}(\cdot|s_h^\ell,a_h^\ell)^\top V_{h+1,\pi_{\mathcal{E}_1}^*}^{\mathcal{E}_1}(\cdot) - P_h^{\mathcal{E}_2}(\cdot|s_h^\ell,a_h^\ell)^\top V_{h+1,\pi_{\mathcal{E}_1}^*}^{\mathcal{E}_2}(\cdot)\right]$$

$$+ \sum_{h=1}^{H} \mathbb{E}_{\pi_{\mathcal{E}_1}^*}^{\mathcal{E}_2}\left[P_h^{\mathcal{E}_2}(\cdot|s_h^\ell,a_h^\ell)^\top \left(V_{h+1,\pi_{\mathcal{E}_1}^*}^{\mathcal{E}_2}(\cdot) - V_{h+1,\pi_{\mathcal{E}_1}^*}^{\mathcal{E}_1}(\cdot)\right)\right]$$

$$\leq \sum_{h=1}^{H} \mathbb{E}_{\pi_{\mathcal{E}_1}^*}^{\mathcal{E}_2}\underbrace{\left[P_h^{\mathcal{E}_1}(\cdot|s_h^\ell,a_h^\ell)^\top V_{h+1,\pi_{\mathcal{E}_1}^*}^{\mathcal{E}_1}(\cdot) - P_h^{\mathcal{E}_2}(\cdot|s_h^\ell,a_h^\ell)^\top V_{h+1,\pi_{\mathcal{E}_1}^*}^{\mathcal{E}_2}(\cdot)\right]}_{I_1}$$

$$+ \sum_{h=1}^{H} \mathbb{E}_{\pi_{\mathcal{E}_1}^*}^{\mathcal{E}_2}\underbrace{\left[\left\|V_{h+1,\pi_{\mathcal{E}_1}^*}^{\mathcal{E}_2}(\cdot) - V_{h+1,\pi_{\mathcal{E}_1}^*}^{\mathcal{E}_1}(\cdot)\right\|_2\right]}_{I_2}.$$

The construction follows the following steps.

- First, we construct a cover for $I_1$. Since the reward is assumed to be bounded by 1, we have $P_h^{\mathcal{E}}(\cdot|s_h^\ell,a_h^\ell)^\top V_{h+1,\pi_{\mathcal{E}}^*}^{\mathcal{E}}(\cdot) \in [0,H]$ for any $\mathcal{E}$. For each $(s,a,h)$, we construct a covering set $\{\mathcal{I}_{sah}^1, \ldots, \mathcal{I}_{sah}^m\}$ for $[0,1]$ where $m = H/\varepsilon$. Thus, each set is of length $\varepsilon$.

- Second, let $\{\mathcal{C}_1, \ldots, \mathcal{C}_M\}$ be an $\varepsilon$-covering of a $S$-dimensional $\ell_2$-ball.

- Third, we construct the partition $\{\Theta_k\}_{k=1}^K$ in the way that $\mathcal{E} \in \Theta_k$ if for any $s,a,h$,

$$\left\langle P_h^{\mathcal{E}}(\cdot|s,a), V_{h+1,\pi_{\mathcal{E}}^*}^{\mathcal{E}}(\cdot)\right\rangle \in \mathcal{I}_{sah}^{k_1}, V_{h+1,\pi_{\mathcal{E}}^*}^{\mathcal{E}}(\cdot)/H \in \mathcal{C}_{k_2},$$

  where $k_1 \in [m], k_2 \in [M]$. The existence of $k_1, k_2$ holds trivially.

Apparently, $\{\Theta_k\}_{k=1}^K$ is a partition of $\Theta$. For any $k \in [K]$ and $\mathcal{E}_1, \mathcal{E}_2 \in \Theta_k$, the following holds for any $s,a,h$,

$$\left|\left\langle P_h^{\mathcal{E}_1}(\cdot|s,a), V_{h+1,\pi_{\mathcal{E}_1}^*}^{\mathcal{E}_1}(\cdot)\right\rangle - \left\langle P_h^{\mathcal{E}_2}(\cdot|s,a), V_{h+1,\pi_{\mathcal{E}_1}^*}^{\mathcal{E}_2}(\cdot)\right\rangle\right| \leq \varepsilon,$$

and
$$\left\| \left( V^{\mathcal{E}_1}_{h+1,\pi^*_{\mathcal{E}_1}}(\cdot) - V^{\mathcal{E}_2}_{h+1,\pi^*_{\mathcal{E}_1}}(\cdot) \right) / H \right\|_2 \le \varepsilon .$$

Therefore, we have constructed a partition $\{\Theta_k\}^K_{k=1}$ over $\Theta$ such that for any $\mathcal{E}_1, \mathcal{E}_2 \in \Theta_k$,
$$V^{\mathcal{E}_1}_{1,\pi^*_{\mathcal{E}_1}}(s_1) - V^{\mathcal{E}_2}_{1,\pi^*_{\mathcal{E}_1}}(s_1) \le \varepsilon ,$$
with the covering number bounded by
$$K \le \left( \frac{H^2}{\varepsilon} \right)^{SAH} + \left( \frac{H^2}{\varepsilon} + 1 \right)^S \le 2(2H^2/\varepsilon)^{SAH} .$$

This ends the proof. $\qquad\square$

## B.4  Proof of Lemma 5.8

*Proof.* We construct a partition $\{\Theta_k\}^K_{k=1}$ over $\Theta^{\mathrm{Lin}}$ such that Eq. (5.1) holds. For any $\mathcal{E}_1, \mathcal{E}_2 \in \Theta_k$, by Lemma D.3,

$$V^{\mathcal{E}_1}_{1,\pi^*_{\mathcal{E}_1}}(s_1) - V^{\mathcal{E}_2}_{1,\pi^*_{\mathcal{E}_1}}(s_1)$$
$$= \sum_{h=1}^{H} \mathbb{E}^{\mathcal{E}_2}_{\pi^*_{\mathcal{E}_1}} \left[ P^{\mathcal{E}_1}_h(\cdot | s^\ell_h, a^\ell_h)^\top V^{\mathcal{E}_1}_{h+1,\pi^*_{\mathcal{E}_1}}(\cdot) - P^{\mathcal{E}_2}_h(\cdot | s^\ell_h, a^\ell_h)^\top V^{\mathcal{E}_1}_{h+1,\pi^*_{\mathcal{E}_1}}(\cdot) \right]$$
$$= \sum_{h=1}^{H} \mathbb{E}^{\mathcal{E}_2}_{\pi^*_{\mathcal{E}_1}} \left[ \phi(s^\ell_h, a^\ell_h)^\top \sum_{s'} V^{\mathcal{E}_1}_{h+1,\pi^*_{\mathcal{E}_1}}(s') \psi^{\mathcal{E}_1}_h(s') - \phi(s^\ell_h, a^\ell_h)^\top \sum_{s'} V^{\mathcal{E}_1}_{h+1,\pi^*_{\mathcal{E}_1}}(s') \psi^{\mathcal{E}_2}_h(s') \right] ,$$

where the last equation is from the definition of linear MDP. Moreover, since the value function is always bounded by $H$, we have

$$V^{\mathcal{E}_1}_{1,\pi^*_{\mathcal{E}_1}}(s_1) - V^{\mathcal{E}_2}_{1,\pi^*_{\mathcal{E}_1}}(s_1) = H \sum_{h=1}^{H} \mathbb{E}^{\mathcal{E}_2}_{\pi^*_{\mathcal{E}_1}} \left[ \phi(s^\ell_h, a^\ell_h)^\top \left( \sum_{s'} \psi^{\mathcal{E}_1}_h(s') - \sum_{s'} \psi^{\mathcal{E}_2}_h(s') \right) \right]$$
$$\le H \sum_{h=1}^{H} \mathbb{E}^{\mathcal{E}_2}_{\pi^*_{\mathcal{E}_1}} \left[ \| \phi(s^\ell_h, a^\ell_h) \|_2 \right] \left\| \sum_{s'} \psi^{\mathcal{E}_1}_h(s') - \sum_{s'} \psi^{\mathcal{E}_2}_h(s') \right\|_2$$
$$\le H \sum_{h=1}^{H} \left\| \sum_{s'} \psi^{\mathcal{E}_1}_h(s') - \sum_{s'} \psi^{\mathcal{E}_2}_h(s') \right\|_2 .$$

From Definition 5.7,
$$\left\| \sum_{s'} \psi^{\mathcal{E}}_h(s') \right\|_2 \le C_\psi .$$

For each $h \in [H]$, let $\{\mathcal{C}^h_1, \dots, \mathcal{C}^h_K\}$ be an $\varepsilon$-covering of a $d$-dimensional $\ell_2$-ball. We construct the partition $\{\Theta_k\}^K_{k=1}$ in the way that $\mathcal{E} \in \Theta_k$ if
$$\frac{1}{C_\psi} \sum_{s'} \psi^{\mathcal{E}}_h(s') \in \mathcal{C}^h_k .$$

Apparently, $\{\Theta_k\}^K_{k=1}$ is a partition of $\Theta$. For any $k \in [K]$ and $\mathcal{E}_1, \mathcal{E}_2 \in \Theta_k$, the following holds
$$\left\| \frac{1}{C_\psi} \sum_{s'} \psi^{\mathcal{E}_1}_h(s') - \frac{1}{C_\psi} \sum_{s'} \psi^{\mathcal{E}_2}_h(s') \right\|_2 \le \varepsilon ,$$
which implies
$$V^{\mathcal{E}_1}_{1,\pi^*_{\mathcal{E}_1}}(s_1) - V^{\mathcal{E}_2}_{1,\pi^*_{\mathcal{E}_1}}(s_1) \le H^2 C_\psi \varepsilon .$$

Letting $\varepsilon' = H^2 C_\psi \varepsilon$, the covering number can be bounded by
$$K \le \left( \frac{H^2 C_\psi}{\varepsilon'} + 1 \right)^{Hd} .$$

This ends the proof. $\qquad\square$

## B.5 Proof of Lemma 5.9

*Proof.* We write $\bar{\mathcal{E}}_\ell^*$ as the MDP consisted by the mean of posterior measure of $\widetilde{\mathcal{E}}_\ell^*$. Noting that $\mathbb{E}_\ell[V_{1,\pi_{\bar{\mathcal{E}}}^*}^{\widetilde{\mathcal{E}}_\ell^*}(s_1^\ell)] = \mathbb{E}_\ell[V_{1,\pi_{\mathrm{TS}}^\ell}^{\widetilde{\mathcal{E}}_\ell^*}(s_1^\ell)]$, we decompose the regret as

$$\mathbb{E}_\ell\left[V_{1,\pi^*}^{\widetilde{\mathcal{E}}_\ell^*}(s_1^\ell) - V_{1,\pi_{\mathrm{TS}}^\ell}^{\widetilde{\mathcal{E}}_\ell^*}(s_1^\ell)\right] = \mathbb{E}_\ell\left[V_{1,\pi^*}^{\widetilde{\mathcal{E}}_\ell^*}(s_1^\ell) - V_{1,\pi^*}^{\widetilde{\mathcal{E}}_\ell}(s_1^\ell)\right]$$

$$= \underbrace{\mathbb{E}_\ell\left[V_{1,\pi^*}^{\widetilde{\mathcal{E}}_\ell^*}(s_1^\ell) - V_{1,\pi^*}^{\bar{\mathcal{E}}_\ell^*}(s_1^\ell)\right]}_{I_1} + \underbrace{\mathbb{E}_\ell\left[V_{1,\pi^*}^{\bar{\mathcal{E}}_\ell^*}(s_1^\ell) - V_{1,\pi^*}^{\widetilde{\mathcal{E}}_\ell}(s_1^\ell)\right]}_{I_2},$$

According to Lemma D.3,

$$I_1 = \mathbb{E}_\ell\left[\sum_{h=1}^H \mathbb{E}_{\pi^*}^{\bar{\mathcal{E}}_\ell}\left[P_h^{\widetilde{\mathcal{E}}_\ell^*}(\cdot|s_h^\ell, a_h^\ell)^\top V_{h+1,\pi^*}^{\widetilde{\mathcal{E}}_\ell^*}(\cdot) - P_h^{\bar{\mathcal{E}}_\ell^*}(\cdot|s_h^\ell, a_h^\ell)^\top V_{h+1,\pi^*}^{\widetilde{\mathcal{E}}_\ell^*}(\cdot)\right]\right].$$

Using the definition of linear MDPs in Definition 5.7,

$$I_1 = \mathbb{E}_\ell\left[\sum_{h=1}^H \mathbb{E}_{\pi^*}^{\bar{\mathcal{E}}_\ell^*}\left[\sum_{s'}\phi(s_h^\ell, a_h^\ell)^\top \psi_h^{\widetilde{\mathcal{E}}_\ell^*}(s')V_{h+1,\pi^*}^{\widetilde{\mathcal{E}}_\ell^*}(s') - \sum_{s'}\phi(s_h^\ell, a_h^\ell)^\top \psi_h^{\bar{\mathcal{E}}_\ell^*}(s')V_{h+1,\pi^*}^{\widetilde{\mathcal{E}}_\ell^*}(s')\right]\right]$$

$$= \mathbb{E}_\ell\left[\sum_{h=1}^H \mathbb{E}_{\pi^*}^{\bar{\mathcal{E}}_\ell^*}\left[\phi(s_h^\ell, a_h^\ell)^\top\right]\sum_{s'}(\psi_h^{\widetilde{\mathcal{E}}_\ell^*}(s') - \psi_h^{\bar{\mathcal{E}}_\ell^*}(s'))V_{h+1,\pi^*}^{\widetilde{\mathcal{E}}_\ell^*}(s')\right].$$

Denoting

$$\Sigma_h = \mathbb{E}_\ell\left[\mathbb{E}_{\pi^*}^{\bar{\mathcal{E}}_\ell^*}\left[\phi(s_h^\ell, a_h^\ell)\right]\mathbb{E}_{\pi^*}^{\bar{\mathcal{E}}_\ell^*}\left[\phi(s_h^\ell, a_h^\ell)^\top\right]\right],$$

we have

$$I_1 = \sum_{h=1}^H \mathbb{E}_\ell\left[\mathbb{E}_{\pi^*}^{\bar{\mathcal{E}}_\ell^*}\left[\phi(s_h^\ell, a_h^\ell)^\top\right]\Sigma_h^{-1/2}\Sigma_h^{1/2}\sum_{s'}(\psi_h^{\widetilde{\mathcal{E}}_\ell^*}(s') - \psi_h^{\bar{\mathcal{E}}_\ell^*}(s'))V_{h+1,\pi^*}^{\widetilde{\mathcal{E}}_\ell^*}(s')\right]$$

By Cauchy–Schwarz inequality, we have

$$I_1 \le \sqrt{\sum_{h=1}^H \mathbb{E}_\ell\left[\left\|\Sigma_h^{1/2}\sum_{s'}(\psi_h^{\widetilde{\mathcal{E}}_\ell^*}(s') - \psi_h^{\bar{\mathcal{E}}_\ell^*}(s'))V_{h+1,\pi^*}^{\widetilde{\mathcal{E}}_\ell^*}(s')\right\|_2^2\right]}\sqrt{\sum_{h=1}^H \mathbb{E}_\ell\left[\left\|\Sigma_h^{-1/2}\mathbb{E}_{\pi^*}^{\bar{\mathcal{E}}_\ell^*}\left[\phi(s_h^\ell, a_h^\ell)\right]\right\|_2^2\right]}.$$

- For the first part,

$$\sum_{h=1}^H \mathbb{E}_\ell\left[\left\|\Sigma_h^{1/2}\sum_{s'}(\psi_h^{\widetilde{\mathcal{E}}_\ell^*}(s') - \psi_h^{\bar{\mathcal{E}}_\ell^*}(s'))V_{h+1,\pi^*}^{\widetilde{\mathcal{E}}_\ell^*}(s')\right\|_2^2\right]$$

$$= \sum_{h=1}^H \mathbb{E}_\ell\left[\left(\mathbb{E}_{\pi^*}^{\bar{\mathcal{E}}_\ell^*}\left[\phi(s_h^\ell, a_h^\ell)^\top\right]\sum_{s'}(\psi_h^{\widetilde{\mathcal{E}}_\ell^*}(s') - \psi_h^{\bar{\mathcal{E}}_\ell^*}(s'))V_{h+1,\pi^*}^{\widetilde{\mathcal{E}}_\ell^*}(s')\right)^2\right]$$

$$= H^2\sum_{h=1}^H \mathbb{E}_\ell\left(\mathbb{E}_{\pi^*}^{\bar{\mathcal{E}}_\ell^*}\left[P_h^{\widetilde{\mathcal{E}}_\ell^*}(\cdot|s_h^\ell, a_h^\ell)^\top V_{h+1,\pi^*}^{\widetilde{\mathcal{E}}_\ell^*}(\cdot)/H - P_h^{\bar{\mathcal{E}}_\ell^*}(\cdot|s_h^\ell, a_h^\ell)^\top V_{h+1,\pi^*}^{\widetilde{\mathcal{E}}_\ell^*}(\cdot)/H\right]\right)^2.$$

Applying Lemma A.1, we have

$$H^2\sum_{h=1}^H \mathbb{E}_\ell\left[\left\|\Sigma_h^{1/2}\sum_{s'}(\psi_h^{\widetilde{\mathcal{E}}_\ell^*}(s') - \psi_h^{\bar{\mathcal{E}}_\ell^*}(s'))V_{h+1,\pi^*}^{\widetilde{\mathcal{E}}_\ell^*}(s')\right\|_2^2\right] = H^2\mathbb{I}_\ell^{\pi^*}\left(\widetilde{\mathcal{E}}_\ell^*; \mathcal{H}_{\ell,H}\right).$$

- For the second part, we can rewrite

$$\mathbb{E}_\ell\left[\left\|\Sigma_h^{-1/2}\mathbb{E}_{\pi^*}^{\bar{\mathcal{E}}_\ell^*}\left[\phi(s_h^\ell, a_h^\ell)\right]\right\|_2^2\right]$$

$$= \left\langle\mathbb{E}_\ell\left[\mathbb{E}_{\pi^*}^{\bar{\mathcal{E}}_\ell^*}\left[\phi(s_h^\ell, a_h^\ell)\right]\mathbb{E}_{\pi^*}^{\bar{\mathcal{E}}_\ell^*}\left[\phi(s_h^\ell, a_h^\ell)^\top\right]\right], \Sigma_h^{-1}\right\rangle = \langle\Sigma_h, \Sigma_h^{-1}\rangle = d.$$

Therefore, putting them together,

$$I_1 \leq \sqrt{H^3 d \mathbb{I}_\ell^{\pi^*} \left( \widetilde{\mathcal{E}}_\ell^*; \mathcal{H}_{\ell,H} \right)}.$$

The derivation of bounding $I_2$ is similar so we omit the detailed proof here. This ends the proof. □

## B.6 Proof of Theorem 5.11

Directly using Lemma 5.1, we can decompose

$$
\begin{aligned}
\mathfrak{BR}_L(\pi_{\mathrm{TS}}) &= \sum_{\ell=1}^{L} \mathbb{E}\left[ \mathbb{E}_\ell \left[ V_{1,\pi^*}^{\mathcal{E}}(s_1^\ell) - V_{1,\pi_{\mathrm{TS}}^\ell}^{\mathcal{E}}(s_1^\ell) \right] \right] \\
&\leq \sum_{\ell=1}^{L} \mathbb{E}\left[ \mathbb{E}_\ell \left[ V_{1,\pi^*}^{\widetilde{\mathcal{E}}_\ell^*}(s_1^\ell) - V_{1,\pi_{\mathrm{TS}}^\ell}^{\widetilde{\mathcal{E}}_\ell^*}(s_1^\ell) \right] \right] + L\varepsilon \\
&\leq \sqrt{ \mathbb{E}\left[ \sum_{\ell=1}^{L} \frac{\left( \mathbb{E}_\ell \left[ V_{1,\pi^*}^{\widetilde{\mathcal{E}}_\ell^*}(s_1^\ell) - V_{1,\pi_{\mathrm{TS}}^\ell}^{\widetilde{\mathcal{E}}_\ell^*}(s_1^\ell) \right] \right)^2}{\mathbb{I}_\ell^{\pi_{\mathrm{TS}}^\ell}(\widetilde{\mathcal{E}}_\ell^*; \mathcal{H}_{\ell,H})} \right]} \sqrt{ \mathbb{E}\left[ \sum_{\ell=1}^{L} \mathbb{I}_\ell^{\pi_{\mathrm{TS}}^\ell}(\widetilde{\mathcal{E}}_\ell^*; \mathcal{H}_{\ell,H}) \right]} + L\varepsilon.
\end{aligned}
$$

By the construction of $\widetilde{\mathcal{E}}_\ell^*$, $\widetilde{\mathcal{E}}_\ell^*$ and $\mathcal{E}$ are independent conditional on $\zeta$. Thus $\widetilde{\mathcal{E}}_\ell^*$ and $\mathcal{H}_{\ell,H}$ are independent under the law $\mathbb{P}_{\ell,\pi_{\mathrm{TS}}^\ell}$ given $\zeta$. By the data processing inequality, we have

$$\mathbb{I}_\ell^{\pi_{\mathrm{TS}}^\ell}(\widetilde{\mathcal{E}}_\ell^*; \mathcal{H}_{\ell,H}) \leq \mathbb{I}_\ell^{\pi_{\mathrm{TS}}^\ell}(\zeta; \mathcal{H}_{\ell,H}).$$

Therefore,

$$
\begin{aligned}
\mathfrak{BR}_L(\pi_{\mathrm{TS}}) &\leq \sqrt{ \mathbb{E}\left[ \sum_{\ell=1}^{L} \frac{\left( \mathbb{E}_\ell \left[ V_{1,\pi^*}^{\widetilde{\mathcal{E}}_\ell^*}(s_1^\ell) - V_{1,\pi_{\mathrm{TS}}^\ell}^{\widetilde{\mathcal{E}}_\ell^*}(s_1^\ell) \right] \right)^2}{\mathbb{I}_\ell^{\pi_{\mathrm{TS}}^\ell}(\widetilde{\mathcal{E}}_\ell^*; \mathcal{H}_{\ell,H})} \right]} \sqrt{ \mathbb{E}\left[ \sum_{\ell=1}^{L} \mathbb{I}_\ell^{\pi_{\mathrm{TS}}^\ell}(\zeta; \mathcal{H}_{\ell,H}) \right]} + L\varepsilon \\
&= \sqrt{ \mathbb{E}\left[ \sum_{\ell=1}^{L} \frac{\left( \mathbb{E}_\ell \left[ V_{1,\pi^*}^{\widetilde{\mathcal{E}}_\ell^*}(s_1^\ell) - V_{1,\pi_{\mathrm{TS}}^\ell}^{\widetilde{\mathcal{E}}_\ell^*}(s_1^\ell) \right] \right)^2}{\mathbb{I}_\ell^{\pi_{\mathrm{TS}}^\ell}(\widetilde{\mathcal{E}}_\ell^*; \mathcal{H}_{\ell,H})} \right]} \sqrt{\mathbb{I}(\zeta; \mathcal{D}_{L+1})} + L\varepsilon.
\end{aligned}
$$

This ends the proof.

# C Proofs of technical lemmas

## C.1 Proof of Lemma A.1

*Proof.* Using the chain rule of mutual information,

$$
\begin{aligned}
\mathbb{I}_\ell^{\pi}(\mathcal{E}; \mathcal{H}_{\ell,H}) &= \sum_{h=1}^{H} \mathbb{E}_\ell \left[ \mathbb{I}_\ell^{\pi}\left( \mathcal{E}; (s_h^\ell, a_h^\ell, r_h^\ell) \big| \mathcal{H}_{\ell,h-1} \right) \right] \\
&= \sum_{h=1}^{H} \mathbb{E}_\ell \left[ \mathbb{I}_\ell^{\pi}\left( \mathcal{E}; s_h^\ell \big| \mathcal{H}_{\ell,h-1} \right) \right] + \sum_{h=1}^{H} \mathbb{E}_\ell \left[ \mathbb{I}_\ell^{\pi}\left( \mathcal{E}; a_h^\ell \big| s_h^\ell, \mathcal{H}_{\ell,h-1} \right) \right] \quad\quad \text{(C.1)} \\
&\quad + \sum_{h=1}^{H} \mathbb{E}_\ell \left[ \mathbb{I}_\ell^{\pi}\left( \mathcal{E}; r_h^\ell \big| s_h^\ell, a_h^\ell, \mathcal{H}_{\ell,h-1} \right) \right].
\end{aligned}
$$

- For the first term in Eq. (C.1), by the definition of conditional mutual information and Markov property, we have

$$\mathbb{I}_\ell^\pi\left(\mathcal{E}; s_h^\ell \middle| \mathcal{H}_{\ell,h-1}\right)$$

$$= \int D_{\mathrm{KL}}\left(\mathbb{P}_{\ell,\pi}\left(s_h^\ell = \cdot | \mathcal{H}_{\ell,h-1}, \mathcal{E}\right) || \mathbb{P}_{\ell,\pi}\left(s_h^\ell = \cdot | \mathcal{H}_{\ell,h-1}\right)\right) \mathrm{d}\mathbb{P}_\ell(\mathcal{E}|\mathcal{H}_{\ell,h-1}) \quad \text{(C.2)}$$

$$= \int D_{\mathrm{KL}}\left(P_h^\mathcal{E}\left(\cdot | s_{h-1}^\ell, a_{h-1}^\ell\right) || \mathbb{P}_{\ell,\pi}\left(s_h^\ell = \cdot | \mathcal{H}_{\ell,h-1}\right)\right) \mathrm{d}\mathbb{P}_\ell(\mathcal{E}|\mathcal{H}_{\ell,h-1}).$$

Since the priors of transition probability kernel are independent over different layers, $\mathbb{P}_\ell(\mathcal{E}|\mathcal{H}_{\ell,h-1}) = \mathbb{P}_\ell(\mathcal{E})$ such that

$$\mathbb{P}_{\ell,\pi}\left(s_h^\ell = \cdot | \mathcal{H}_{\ell,h-1}\right) = \int \mathbb{P}_{\ell,\pi}\left(s_h^\ell = \cdot | \mathcal{H}_{\ell,h-1}, \mathcal{E}\right) \mathrm{d}\mathbb{P}_\ell(\mathcal{E}|\mathcal{H}_{\ell,h-1})$$

$$= \int P_h^\mathcal{E}(\cdot | s_{h-1}^\ell, a_{h-1}^\ell) \mathrm{d}\mathbb{P}_\ell(\mathcal{E}|\mathcal{H}_{\ell,h-1}) \quad \text{(C.3)}$$

$$= \int P_h^\mathcal{E}(\cdot | s_{h-1}^\ell, a_{h-1}^\ell) \mathrm{d}\mathbb{P}_\ell(\mathcal{E})$$

$$= P_h^{\bar{\mathcal{E}}_\ell}\left(\cdot | s_{h-1}^\ell, a_{h-1}^\ell\right),$$

where the last equation is by the definition of probability kernel $P_h^{\bar{\mathcal{E}}_\ell}$. Combining Eqs. (C.2) and (C.3) together,

$$\mathbb{I}_\ell^\pi\left(\mathcal{E}; s_h^\ell \middle| \mathcal{H}_{\ell,h-1}\right) = \int D_{\mathrm{KL}}\left(P_h^\mathcal{E}(\cdot | s_{h-1}^\ell, a_{h-1}^\ell) || P_h^{\bar{\mathcal{E}}_\ell}(\cdot | s_{h-1}^\ell, a_{h-1}^\ell)\right) \mathrm{d}\mathbb{P}_\ell(\mathcal{E}).$$

Therefore,

$$\mathbb{E}_\ell\left[\mathbb{I}_\ell^\pi\left(\mathcal{E}; s_h^\ell \middle| \mathcal{H}_{\ell,h-1}\right)\right]$$

$$= \mathbb{E}_\ell\left[\int D_{\mathrm{KL}}\left(P_h^\mathcal{E}(\cdot | s_{h-1}^\ell, a_{h-1}^\ell) || P_h^{\bar{\mathcal{E}}_\ell}(\cdot | s_{h-1}^\ell, a_{h-1}^\ell)\right) \mathrm{d}\mathbb{P}_\ell(\mathcal{E})\right]$$

$$= \sum_{(s,a)} \mathbb{P}_{\ell,\pi}(s_{h-1}^\ell = s, a_{h-1}^\ell = a) \int D_{\mathrm{KL}}\left(P_h^\mathcal{E}(\cdot | s, a) || P_h^{\bar{\mathcal{E}}_\ell}(\cdot | s, a)\right) \mathrm{d}\mathbb{P}_\ell(\mathcal{E})$$

$$= \sum_{(s,a)} \int \mathbb{P}_{\ell,\pi}(s_{h-1}^\ell = s, a_{h-1}^\ell = a | \mathcal{E}) \mathrm{d}\mathbb{P}_\ell(\mathcal{E}) \int D_{\mathrm{KL}}\left(P_h^\mathcal{E}(\cdot | s, a) || P_h^{\bar{\mathcal{E}}_\ell}(\cdot | s, a)\right) \mathrm{d}\mathbb{P}_\ell(\mathcal{E}).$$

Using the linearity of expectation and the independence of priors over different layers, we can show

$$\int \mathbb{P}_{\ell,\pi}(s_{h-1}^\ell = s, a_{h-1}^\ell = a | \mathcal{E}) \mathrm{d}\mathbb{P}_\ell(\mathcal{E}) = \mathbb{P}_\pi^{\bar{\mathcal{E}}_\ell}(s_{h-1}^\ell = s, a_{h-1}^\ell = a).$$

This implies

$$\mathbb{E}_\ell\left[\mathbb{I}_\ell^\pi\left(\mathcal{E}; s_h^\ell \middle| \mathcal{H}_{\ell,h-1}\right)\right]$$

$$= \sum_{(s,a)} \mathbb{P}_\pi^{\bar{\mathcal{E}}_\ell}(s_{h-1}^\ell = s, a_{h-1}^\ell = a) \int D_{\mathrm{KL}}\left(P_h^\mathcal{E}(\cdot | s, a) || P_h^{\bar{\mathcal{E}}}(\cdot | s, a)\right) \mathrm{d}\mathbb{P}_\ell(\mathcal{E})$$

$$= \int \mathbb{E}_\pi^{\bar{\mathcal{E}}_\ell}\left[D_{\mathrm{KL}}\left(P_h^\mathcal{E}(\cdot | s_{h-1}^\ell, a_{h-1}^\ell) || P_h^{\bar{\mathcal{E}}_\ell}(\cdot | s_{h-1}^\ell, a_{h-1}^\ell)\right)\right] \mathrm{d}\mathbb{P}_\ell(\mathcal{E})$$

$$= \mathbb{E}_\ell\left[\mathbb{E}_\pi^{\bar{\mathcal{E}}_\ell}\left[D_{\mathrm{KL}}\left(P_h^\mathcal{E}(\cdot | s_{h-1}^\ell, a_{h-1}^\ell) || P_h^{\bar{\mathcal{E}}_\ell}(\cdot | s_{h-1}^\ell, a_{h-1}^\ell)\right)\right]\right],$$

where $\mathbb{E}_\pi^{\bar{\mathcal{E}}_\ell}$ is taken with respect to $s_{h-1}^\ell, a_{h-1}^\ell$ and $\mathbb{E}_\ell$ is taken with respect to $\mathcal{E}$.

- For the second term of Eq. (C.1), we have

$$\mathbb{I}_\ell^\pi\left(\mathcal{E}; a_h^\ell \middle| s_h^\ell, \mathcal{H}_{\ell,h-1}\right)$$

$$= \int D_{\mathrm{KL}}\left(\mathbb{P}_{\ell,\pi}\left(a_h^\ell = \cdot | \mathcal{H}_{\ell,h-1}, s_h^\ell, \mathcal{E}\right) || \mathbb{P}_{\ell,\pi}\left(a_h^\ell = \cdot | s_h^\ell, \mathcal{H}_{\ell,h-1}\right)\right) \mathrm{d}\mathbb{P}_\ell(\mathcal{E}).$$

When $\mathcal{H}_{\ell,h-1}^\pi$ is fixed, both sides of the KL term are equal to $\pi(\cdot|s_h^\ell)$ and thus $\mathbb{I}_\ell^\pi(\mathcal{E}; a_h^\ell|s_h^\ell, \mathcal{H}_{\ell,h-1}) = 0$.

- For the third term of Eq. (C.1), since the reward function is deterministic and known, we have

$$\mathbb{I}_\ell^\pi\left(\mathcal{E}; r_h^\ell|s_h^\ell, a_h^\ell, \mathcal{H}_{\ell,h-1}^\pi\right) = 0\,.$$

This suffices to show

$$\sum_{h=1}^H \mathbb{E}_\ell \mathbb{E}_{\pi_{\mathrm{TS}}^\ell}^{\bar{\mathcal{E}}_\ell}\left[ D_{\mathrm{KL}}\left(P_h^\mathcal{E}(\cdot|s_h,a_h)||P_h^{\bar{\mathcal{E}}_\ell}(\cdot|s_h,a_h)\right)\right] = \mathbb{I}_\ell^\pi\left(\mathcal{E}; \mathcal{H}_{\ell,H}\right)\,. \tag{C.4}$$

This ends the proof.

$\square$

## D  Supporting lemmas

**Lemma D.1** (Lemma 1 in Dong and Van Roy [2018]). *Let $\{a_i\}_{i=1}^N$ and $\{b_i\}_{i=1}^N$ be two sequences of real numbers, where $N < \infty$. Let $\{p_i\}_{i=1}^N$ be such that $p_i \geq 0$ for all $i$ and $\sum_{i=1}^N p_i = 1$. Then there exists indices $j,k \in [N]$ and $r \in [0,1]$ such that*

$$ra_j + (1-r)a_k \leq \sum_{i=1}^N a_i p_i, \quad rb_j + (1-r)b_k \leq \sum_{i=1}^L b_i p_i\,.$$

**Lemma D.2** (Fact 9 in Russo and Van Roy [2014]). *For any distribution $P$ and $Q$ such that $P$ is absolutely continuous with respect to $Q$, any random variable $X : \Omega \to \mathcal{X}$ and any $g : \mathcal{X} \to \mathbb{R}$ such that $\sup g - \inf g \leq 1$, we have*

$$\mathbb{E}_P[g(x)] - \mathbb{E}_Q[g(x)] \leq \sqrt{\frac{1}{2} D_{\mathrm{KL}}(P||Q)}\,,$$

*which is a variant of Pinsker's inequality.*

**Lemma D.3.** *For any two environments $\mathcal{E}, \mathcal{E}'$, any policy $\pi$ and a fixed set of reward functions $\{r_h\}_{h=1}^H$, we have*

$$V_{1,\pi}^\mathcal{E}(s_1) - V_{1,\pi}^{\mathcal{E}'}(s_1) = \sum_{h=1}^H \mathbb{E}_\pi^{\mathcal{E}'}\left[\mathbb{E}_{s'\sim P_h^\mathcal{E}(\cdot|s_h,a_h)}[V_{h+1,\pi}^\mathcal{E}(s')] - \mathbb{E}_{s'\sim P_h^{\mathcal{E}'}(\cdot|s_h,a_h)}[V_{h+1,\pi}^\mathcal{E}(s')]\right]\,,$$

*where we define $V_{H+1,\pi^*}^\mathcal{E}(\cdot) = 0$ and the outer expectation $\mathbb{E}_\pi^{\mathcal{E}'}$ is with respect to $s_h, a_h$.*

*Proof.* Similar proofs can be found in Osband et al. [2013], Foster et al. [2021]. For the self-completeness, we include a full proof here. First, we realize

$$\sum_{h=1}^H \mathbb{E}_\pi^{\mathcal{E}'}\left[Q_{h,\pi}^\mathcal{E}(s_h,a_h) - r_h(s_h,a_h) - V_{h+1,\pi}^\mathcal{E}(s_{h+1})\right]$$

$$= \sum_{h=1}^H \mathbb{E}_\pi^{\mathcal{E}'}\left[Q_{h,\pi}^\mathcal{E}(s_h,a_h) - V_{h+1,\pi}^\mathcal{E}(s_{h+1})\right] - \sum_{h=1}^H \mathbb{E}_\pi^{\mathcal{E}'}\left[r_h(s_h,a_h)\right] \tag{D.1}$$

$$= \sum_{h=1}^H \mathbb{E}_\pi^{\mathcal{E}'}\left[Q_{h,\pi}^\mathcal{E}(s_h,a_h) - V_{h+1,\pi}^\mathcal{E}(s_{h+1})\right] - V_{1,\pi}^{\mathcal{E}'}(s_1)\,.$$

Since $V_{h,\pi}^\mathcal{E}(s) = \mathbb{E}_{a\sim\pi_h(\cdot|s)}[Q_{h,\pi}^\mathcal{E}(s,a)]$, we have

$$\sum_{h=1}^H \mathbb{E}_\pi^{\mathcal{E}'}\left[Q_{h,\pi}^\mathcal{E}(s_h,a_h) - V_{h+1,\pi}^\mathcal{E}(s_{h+1})\right] = \sum_{h=1}^H \mathbb{E}_\pi^{\mathcal{E}'}\left[V_{h,\pi}^\mathcal{E}(s_h) - V_{h+1,\pi}^\mathcal{E}(s_{h+1})\right]$$

$$= \mathbb{E}_\pi^{\mathcal{E}'}\left[V_{1,\pi}^\mathcal{E}(s_1)\right] = V_{1,\pi}^\mathcal{E}(s_1)\,. \tag{D.2}$$

Using the Bellman equation, we have

$$V_{1,\pi}^{\mathcal{E}}(s_1) - V_{1,\pi}^{\mathcal{E}'}(s_1)$$
$$= \sum_{h=1}^{H} \mathbb{E}_{\pi}^{\mathcal{E}'} \left[ \mathbb{E}_{s' \sim P_h^{\mathcal{E}}(\cdot | s_h, a_h)} [V_{h+1,\pi}^{\mathcal{E}}(s')] - \mathbb{E}_{s' \sim P_h^{\mathcal{E}'}(\cdot | s_h, a_h)} [V_{h+1,\pi}^{\mathcal{E}}(s')] \right].$$

This ends the proof. $\square$