# OpenReview forum: "Regret Bounds for Information-Directed Reinforcement Learning"
_NeurIPS.cc/2022/Conference — NeurIPS 2022 Accept_

### Official Review · Reviewer_dRgt · 2022-06-24

**Rating:** 7
**Confidence:** 3
**Soundness:** 3 good
**Presentation:** 3 good
**Contribution:** 3 good

**Summary:**

In this paper, the authors studied the provable efficient Information-Directed Sampling (IDS) methods in MDP setting. They first proposed vanilla-IDS and then derived a prior-free Bayesian regret bound for it. After that, for the sake of computational efficiency, they proposed another variant called regularized-IDS. Besides, they improved the regret bound by learning a surrogate environment. Beyond the tabular setting, they also extended their results to linear MDP.

**Questions:**

Please check the **Weakness** section above.

**Limitations:**

N.A.

**Strengths And Weaknesses:**

### Strengths

This paper has an important contribution to understanding IDS methods in the MDP setting. The algorithm and analysis look novel and interesting. The paper writing looks good to me.


### Weakness

I only have a small issue about the rigorousness in the discussion related to $\Gamma^*$ in Sec. 3.1 and related proof.

It seems to me that $\Gamma_l(\pi^l_{IDS})$ is not a constant across $l=1,2,...,L$, while $\Gamma^*$ is defined to be the worst-case information ratio and upper bounds $\Gamma_l(\pi^l_{IDS})$ for all $l\in[L]$. As a result, there might exists some $\bar{l},\tilde{l} \in [L]$, such that $\Gamma^*$ is attained at $\bar{l}$, but at $\tilde{l}$, we have $\Gamma_{\tilde{l}}(\pi^\tilde{l}_{IDS}) < \Gamma^*$.

Therefore, although we always have $\Gamma_l(\pi^l_{IDS}) \leq \Gamma_l(\pi^l_{TS})$, it is possible that $\Gamma_l(\pi^l_{IDS}) \leq \Gamma_l(\pi^l_{TS}) < \Gamma^*$ when $l=\tilde{l}$. As a result, I think the argument $\Gamma^* \leq \Gamma_l(\pi^l_{TS})$ (Line 482 in the proof of Lem. 3.2) is not correct (but I guess one can recover the same regret upper bound without introducing $\Gamma^*$ and therefore there will be no such issue).

If the authors can fix the issue I mentioned above, I would like to increase my score correspondingly.

---

> ### Author Response · Authors · 2022-08-02
> **Point by point response**
>
> Thanks a lot for acknowledging our contribution! We would like to respond to comments point by point.
>
> 1. **“rigorousness in the discussion related to Γ∗in Sec. 3.1 and related proof.”**
>
> Thanks a lot for the careful reading! We completely agree with your argument. We feel there is a typo here. In Line 482 of the proof of Lemma 3.2, the argument should be
> $$
> \Gamma_*\leq \max_{\ell\in[L]} \Gamma_\ell(\pi\_{TS}^\ell)
> $$
> since the bound of $\Gamma_\ell$ is independent of $\ell$. We hope this could clarify your question.

---

> > ### Comment · Reviewer_dRgt · 2022-08-07
> > **Post Rebuttal**
> >
> > Thanks for the response. I raise my score correspondingly.

---

> > > ### Author Response · Authors · 2022-08-08
> > > **Thanks a lot!**
> > >
> > > Thanks a lot for raising the score!

---

### Official Review · Reviewer_hin3 · 2022-07-09

**Rating:** 6
**Confidence:** 3
**Soundness:** 4 excellent
**Presentation:** 3 good
**Contribution:** 2 fair

**Summary:**

This paper presents general guarantees for information directed sampling in MDPs. As it stood, prior work had only understood Thompson-Sampling inspired approaches in frequentist settings, or provided bounds for specific priors, but this is the first work to analyze proper IDS for MDPs with no restrictions on the prior.

**Questions:**

Do the authors conjecture that the suboptimality of their regret is a limitation of the analysis, or the algorithm? To they have any analysis or experimental evidence to shed light on this? Moreover, have the authors thought about what a more computationally efficient algorithm which uses the MDP cover would look like?

**Limitations:**

As noted, regret bounds are suboptimal, refined bounds are not computationally efficient.

**Strengths And Weaknesses:**

Strengths: the bounds in this paper apply to general priors, some of the information bounds based on the method of mixtures may be of independent interest, a regularized variant of IDS can be implemented efficiently given access to a natural sampling oracle. In addition, the paper is generally well written and well explained, despite a couple of minor grammatical issues. Authors do a great job explaining what the essential ingredients are of their proofs.

The refinements due to rate distortion theory were also a nice addition.

Weaknesses: I should preface this by saying that I am not an expert on Bayesian regret bounds; hence it is hard for me to gauge the technical contribution of this paper. However, it does seem that the techniques and arguments are rather standard, and I think it would be useful for the authors do explain not just the *results* derived in prior works, but to give a sense of how common (or unique) their techniques are in comparison to the rest of the Bayesian regret community.

In addition, it seems that the bounds here do not match what is attainable in the (harder) frequentist setting. This makes me wonder - either is (a) the analysis loose, or (b) can one derive lower bounds to show that IDS (without modification) necessarily suffers this worse sampling complexity? Even some numerical experiments demonstrating scaling with S would be illustrative here.

Another weakness is that the sharper guarantees required computing an explicit cover, which is computationally prohibitive. I would have been more excited if the refined regret were attainable with computationally efficient algorithms.

---

> ### Author Response · Authors · 2022-08-02
> **Point by point response**
>
> Thanks for your thoughtful review. We would like to respond to comments point by point.
>
> 1. **“However, it does seem that the techniques and arguments are rather standard, and I think it would be useful for the authors do explain not just the results derived in prior works, but to give a sense of how common (or unique) their techniques are in comparison to the rest of the Bayesian regret community.”**
>
> Thanks for your suggestion! In literature, there are two ways to prove Bayesian regret bounds:
> - The first one is to introduce confidence sets. But this **cannot** be used to analyze IDS to the best of our knowledge.
> - The second one is to use information-theoretical analysis but almost all the analysis is limited to bandits setting (Russo and Ben, 2014). Extending such techniques to the MDP case is highly non-trivial since we need to model the randomness from the transition dynamic. As commented by Reviewer uxAT, although similar techniques appear in different literature, our work is **the first one** to use information-theoretic analysis to analyze Bayesian regret in MDPs.
>
> We also would like to highlight one of key our technical contributions is Lemma A.1.
>
> **Lemma A.1.** For environment $\mathcal E$ and its corresponding mean of posterior measure $\bar {\mathcal E}\_\ell$, the following holds for any policy $\pi$,
> \begin{equation*}
>    \sum_{h=1}^H \mathbb E_{\ell}\mathbb E_{\pi}^{\bar{\mathcal E}\_\ell}\left[D_{KL}\left(P_h^{\mathcal E}(\cdot|s_h^\ell,a_h^\ell)||P_h^{\bar{\mathcal E}\_\ell}(\cdot|s_h^\ell,a_h^\ell)\right)\right]= \mathbb I_\ell^{\pi}\left(\mathcal E; \mathcal
>  H_{\ell, H}\right).
> \end{equation*}
>
> By exploiting the property of independent priors, we can relate the mutual information with the KL w.r.t the mean of posterior measure. We believe this is **the first of this kind** of results in literature and this lemma is critical to bound the Bayesian regret of IDS and to derive the computational-efficient version (regularized-IDS).
>
> 2. **This makes me wonder - either is (a) the analysis loose, or (b) can one derive lower bounds to show that IDS (without modification) necessarily suffers this worse sampling complexity?”**
>
> Thanks for your question. For vanilla IDS (Section 3.1) where the agent needs to learn exactly the whole transition dynamic, we conjecture that this algorithm cannot achieve optimal regret bound in the tabular case since learning every part of the whole dynamic is redundant to learn the optimal policy.
>
> For surrogate-IDS, we believe its regret bound could be tightened and should be optimal. At this moment, the loose part comes from the bound of information-ratio. When the prior is specified to Dirichlet prior, Lu and Van Roy (2019) has shown that the upper bound of information-ratio can be independent of S (in contrast, our current upper bound is SA but holds for any prior) which leads to an optimal regret bound in our case. It will be an important future work to tighten the upper bound of the information-ratio for any prior.
>
> 3. **“Another weakness is that the sharper guarantees required computing an explicit cover, which is computationally prohibitive. I would have been more excited if the refined regret were attainable with computationally efficient algorithms.”**
>
> This is a very good question and we do not have an immediate answer here. Another possible way is to directly learn the optimal value function or optimal policy rather than computing an explicit cover for the environment. We believe this is an interesting future work.

---

> > ### Comment · Reviewer_hin3 · 2022-08-03
> > **I appreciate the rebuttal.**
> >
> > Thank you for the detailed rebuttal. I do agree that your work is the first to extend IDS to MDPs. My concern is, "are any of the conceptual steps required to do so that challenging?" For example, the tensorization of the KL divergence across time steps, which seems to be at the heart of Lemma A.1, seems to have been leveraged in frequentist settings, see e.g. : https://arxiv.org/abs/1806.00775
> >
> > I am not saying no novelty is present. I am just unsure about the extent to which the novelty and challenge makes me overly enthusiastic about acceptance. However, I do think that the simplicity of the approach is a merit, and the writing makes me view the work favorably. I'll raise my score to a 6.

---

> > > ### Author Response · Authors · 2022-08-08
> > > **Thanks a lot!**
> > >
> > > Thanks a lot for appreciating our contribution!

---

### Official Review · Reviewer_Dwor · 2022-07-12

**Rating:** 6
**Confidence:** 3
**Soundness:** 3 good
**Presentation:** 2 fair
**Contribution:** 2 fair

**Summary:**

This paper studies information-directed sampling (IDS) for Markov decision processes. In particular, the authors prove the Bayesian regret bound of IDS for finite horizon tabular MDPs and linear MDPs.


**Questions:**

Since $\mathcal{S}, \mathcal{A}$ and $r_h$ are assumed to be known and deterministic, the expectation over the environment $\mathcal{E}$ is just the expectation over the prior of the transition probabilities?


**Limitations:**

Line 51: it would be more convincing to include some cases where the best UCB-type algorithms are sub-optimal.

One potential drawback of the paper is its lack of a specific example for calculating the information ratio and the sample complexity for estimating it, which makes it hard to understand the advantage of using IDS policy in practice.

Line 177: conditionar => conditional


**Strengths And Weaknesses:**

The writing of the paper is clear and the proofs seem to be sound. While I appreciate the comparison of the regret bound with other methods in the literature, I do not think it is proper to compare the Bayesian regret bound derived in this paper with the frequentist regret bound in other papers. Special comments should be carefully made around any of these remarks.

---

> ### Author Response · Authors · 2022-08-02
> **Point by point response**
>
> Thanks for your thoughtful review! We would like to respond to comments point by point.
>
> 1. **“While I appreciate the comparison of the regret bound with other methods in the literature, I do not think it is proper to compare the Bayesian regret bound derived in this paper with the frequentist regret bound in other papers. Special comments should be carefully made around any of these remarks.”**
>
> Thanks for your suggestion! We completely agree on this. In the revision, when comparing the regret upper bounds, we have explicitly mentioned if the upper bound is Bayesian regret or frequentist regret and added the comments that this is not an apples-to-apples comparison.
>
> 2. **“Since S,A and rh are assumed to be known and deterministic, the expectation over the environment E is just the expectation over the prior of the transition probabilities?”**
>
> Yes, this is true. The extension to unknown and stochastic reward functions is straightforward.
>
> 3. **“Line 51: it would be more convincing to include some cases where the best UCB-type algorithms are sub-optimal.”**
>
> Thanks for the suggestion. In Section 4 of the work “Information Directed Sampling for Sparse Linear Bandits, NeurIPS 2021”, the author has proven that any UCB-type algorithm could be sub-optimal in terms of minimax regret for sparse linear bandits. We have included this case in the revision.
>
> 4. **“One potential drawback of the paper is its lack of a specific example for calculating the information ratio and the sample complexity for estimating it, which makes it hard to understand the advantage of using IDS policy in practice.”**
>
> Thank you very much for your comments! In the revision, we have included a detailed algorithm box for implementing regualized-IDS for tabular MDPs in Appendix A.
>
> In practice, it is usually expensive to directly calculate the KL-distance. Following the idea from Russo and Ben (2018) (Learning to Optimize via Information-Directed Sampling), we could lower bound the KL-distance by the variance as follows. By Pinsker's inequality,
>
> \begin{equation}
> \begin{split}
>  \int D_{KL}\left(P_h^{\mathcal E}(\cdot|s,a)||P_{h}^{\bar{\mathcal E}_\ell}(\cdot|s,a)\right)d \mathbb P_\ell(\mathcal E)&\geq \int \left\|P_h^{\mathcal E}(\cdot|s,a)-P_h^{\bar{\mathcal E}_\ell}(\cdot|s,a)\right\|_2^2 d \mathbb P_\ell(\mathcal E)\\\\
> &\geq \int\sum\_{s'} \left(P_h^{\mathcal E}(s'|s,a)-P_h^{\bar{\mathcal E}_\ell}(s'|s,a)\right)^2 d \mathbb P_\ell(\mathcal E)\\\\
> &=\sum\_{s'}\text{Var}\left(P_h^{\mathcal E}(s'|s,a)\right).
> \end{split}
> \end{equation}
>
> Then the augmented reward function in terms of variance terms is:
>  \begin{equation}
>  r'_h(s,a) =  r_h(s,a)+\lambda\sum\_{s'}\text{Var}\left(P_h^{\mathcal E}(s'|s,a)\right).
>  \end{equation}
> With independent Dirichlet prior, both $\text{Var}\left(P_h^{\mathcal E}(s'|s,a)\right)$ and $P_h^{\bar{\mathcal E}_\ell}(\cdot|s,a)$ have the closed form. We can also prove that this version of regularized-IDS enjoys the same regret bound.
>
> **In Appendix A of the revision**, we conducted preliminary experiments using the RiverSwim and stochastic chain MDP environment and compared the empirical performance of posterior sampling for reinforcement learning (PSRL) and regularized-IDS. Both algorithms use the same Dirichlet priors. We use the theoretical suggested regularization parameter which is chosen as $\lambda\times\sqrt{L}$, where $L$ is the number of total episodes. For a proper choice of tuning parameters $\lambda$, regularized-IDS can outperform PSRL as illustrated below (cumulative empirical regrets over 10K episodes). This confirms the advantage of using IDS policy in practice.
>
> | Env | Regualized IDS, lambda=0.1 | Regualized IDS, lambda=0.5 | PSRL |
> | ----------- | ----------- | ----------- | ----------- |
> | RiverSwim | 136.54 | 369.23 | 229.67
> | Chain MDP | 35.97 | 66.55 | 73.71
>
>
> 5. **“Line 177: conditionar => conditional”**
>
> Thanks! We have corrected this in the revision.

---

> > ### Comment · Reviewer_Dwor · 2022-08-08
> > **Thank you for your detailed response.**
> >
> > I appreciate the authors's response, which address all of my questions. Sorry for the late reply. And I am actually satisfied with the rebuttal and especially the newly added algorithm and its implementaton. I have raised my score for this work.

---

> ### Author Response · Authors · 2022-08-08
> **Any further question**
>
> Dear reviewer,
>
> Please let us know if you have further questions about our response. Thanks a lot.

---

### Official Review · Reviewer_uxAT · 2022-07-12

**Rating:** 7
**Confidence:** 4
**Soundness:** 3 good
**Presentation:** 3 good
**Contribution:** 3 good

**Summary:**

This paper provides the first information-directed sampling (IDS) algorithm with theoretical guarantees for the learning in episodic MDP in the prior free setting. The assumptions are the following: the reward function is deterministic and known to the learner and the transition probabilities in the MDP are unknown and are sampled from a known prior distribution before the first episode begins. Authors considers two types of setups: all presented algorithms works in a tabular setting and some results also extended to work in the linear MDPs. The performance of the learner is measured by a Bayesian regret.

**Questions:**

In some places the definition of variables are omitted, please check it.
- Definition of \bar{\Epsilon}_l is recursive
-  zeta in the proof of B.1 is undefined
-  the way of defining \pi_{TS}^l is confusing as this policy is only used in the proof and no presented algorithms use it to compute the actual policy.

Minor remarks

- for the clarity, it has to be mentioned in the preliminaries that the prior is assumed to be known to the learner
- Partition should depend on \epsilon
- In equation (3.3), \pi is missing from I_{\ell}


**Limitations:**

The main limitation of the proposed algorithms is that they are not computationally efficient.

**Strengths And Weaknesses:**

This works adapts the idea of IDS to the setting of learning in the MDP and the authors present three algorithms to tackle this problem. For the first algorithm proposed, Vanilla IDS, the idea is to introduce a notion of the “environment”, which hides in it all the randomness of the unknown parameters of MDP’s transitions and define the information ratio for a policy \pi as the ratio between the square of expected difference of the value function of optimal policy and the value function of policy \pi, divided by the information gain of the “environment” variable and a the history of episode ℓ up to layer h produced by a policy \pi, all conditioned on the history. To find a \pi, that achieves the minimum information ratio, the learned has to optimize over the full policy space, which is a computationally costly. The analysis is simple and borrows the tricks from the literature, as the decomposition of regret based on the marginal posterior distribution of  “environment” (line 145) and a trick with the ratio of occupancy measures (Lemma D.3), but all together it gives the first regret bound of this kind.
Next, authors propose Regularized-IDS algorithm, where instead of computing the ratio, authors propose to compute the sum of the arguments of Vanilla IDS. The result of this chapter is that the
 Regularized-IDS can be efficiently computed using the samples from the posterior which gives the augmented MDP and has the same regret bound as Vanilla IDS.
Finally, the author improve the regret bound of Regularized-IDS  and Vanilla IDS, which they show can be achievable by Surrogate-IDS algorithm. The idea of this algorithm is to construct a surrogate environment, which would be an \epsilon approximation of the true “environment” variable and then compute the information ratio which would be computed over this approximated environment. This algorithm is not computationally efficient, but it improves the dependence of the regret bound on S. Also, the discretisation approach allows to extend the results obtained for episodic MDP to linear MDP, as the number of the set in the partition of the environment space does grow as the covering number of the bounded set in R^d.


I find it especially interesting how similar techniques works in the analysis of this paper and [Foster et.al 2021],  since it give another evidence that the decision-estimated coefficient is related to the information ratio.

---

> ### Author Response · Authors · 2022-08-02
> **Point by point response**
>
> Thanks for your thoughtful review! We would like to respond to comments point by point.
>
> 1. **“Definition of \bar{\Epsilon}_l is recursive”**
>
> Thanks for pointing this out! We have modified the definition to
>
> "We define $\bar{\mathcal E}_{\ell}$ as the mean MDP where for each state-action pair (s,a), $P\_{h}^{\bar{\mathcal E}\_{\ell}}(\cdot|s,a)=\mathbb E_\ell[P_h^{\mathcal E}(\cdot|s,a)]$ is the mean of posterior measure."
>
> 2. **“zeta in the proof of B.1 is undefined”**
>
>
> Thanks for asking. We have defined $\zeta$ in Line 239 in the main paper:
>
> “Let $\zeta$ be a discrete random variable taking values in $\{1, . . . , K\}$ that indicates the region $\mathcal E$ lies such that $\zeta=k$ if and only if $\mathcal E\in\Theta_k$.“
>
> We have restated the definition in the proof of B.1 in the revision.
>
> 3. **“the way of defining \pi_{TS}^l is confusing as this policy is only used in the proof and no presented algorithms use it to compute the actual policy.”**
>
>
> We would like to mention that we also present the Bayesian regret bound for $\pi_{TS}^\ell$ in Section 4.4.
>
> 4. **“for the clarity, it has to be mentioned in the preliminaries that the prior is assumed to be known to the learner”**
>
> Thanks! We have mentioned this explicitly in the revision.
>
> 5. **“Partition should depend on \epsilon”**
>
> Thanks! We have made the partition depending on $\epsilon$ in the revision.
>
> 6. **“In equation (3.3), $\pi$ is missing from $I_{\ell}$”**
>
> Thanks! We have added $\pi$ in the revision.

---

> > ### Comment · Reviewer_uxAT · 2022-08-08
> > **rebuttal reply**
> >
> > Thank you for addressing my comments.

---

### Author Response · Authors · 2022-08-02
**General response and revision**

Dear reviewers,

Thanks a lot for your valuable reviews and suggestion! We have revised our draft according to your comments and added a detailed implementation and some preliminary experiments for regularized-IDS (Appendix A).

Best,

Authors

---

### Meta-Review · Area_Chair_tcU4 · 2022-08-23

**Recommendation:** Accept
**Confidence:** Certain

**Metareview:**

This paper has been well-received by the reviewers already in the initial round, and the reviewers were all happy with the authors' responses. The updates already made to the manuscript clearly showed the commitment of the authors to take all the reviewers' comments into account for the final version. After some discussion, all reviewers agreed that the paper should be accepted for publication at NeurIPS 2022. I encourage the authors to finalize the promised changes for the camera-ready version, and in particular complete the preliminary experimental section provided in the revision.

**Award:**

No

---

### Decision · Program_Chairs · 2022-09-14

Accept